   

THE EMBO JOURNAL

# YIPF3 and YIPF4 regulate autophagic turnover of the Golgi apparatus

Shinri Kitta [1], Tatsuya Kaminishi [1,2], Momoko Higashi[3], Takayuki Shima[1], Kohei Nishino[4], Nobuhiro Nakamura[5], Hidetaka Kosako [4], Tamotsu Yoshimori [1,2,3 ✉] & Akiko Kuma [1,6 ✉]

## Abstract

The degradation of organelles by autophagy is essential for cellular homeostasis. The Golgi apparatus has recently been demonstrated to be degraded by autophagy, but little is known about how the Golgi is recognized by the forming autophagosome. Using quantitative proteomic analysis and two novel Golgiphagy reporter systems, we found that the five-pass transmembrane Golgi-resident proteins YIPF3 and YIPF4 constitute a Golgiphagy receptor. The interaction of this complex with LC3B, GABARAP, and GABARAPL1 is dependent on a LIR motif within YIPF3 and putative phosphorylation sites immediately upstream; the stability of the complex is governed by YIPF4. Expression of a YIPF3 protein containing a mutated LIR motif caused an elongated Golgi morphology, indicating the importance of Golgi turnover via selective autophagy. The reporter assays reported here may be readily adapted to different experimental contexts to help deepen our understanding of Golgiphagy.

**Keywords** Golgiphagy; Selective Autophagy; Golgi Turnover; Organelle Degradation; Autophagy Receptor
**Subject Categories** Autophagy & Cell Death; Organelles

## Introduction

The turnover of organelles is crucial for cellular homeostasis, especially when they are damaged. One of the ways of removing such organelles is macroautophagy (hereafter, autophagy). Autophagy is an evolutionarily conserved cytoplasmic degradation system in which autophagosomes engulf cytoplasmic materials, including organelles, and fuse with lysosomes to degrade them. In this process, several autophagy-related (*ATG*) proteins orchestrate autophagosome formation and maturation (Mizushima et al, 2011; Nakatogawa, 2020). ATG proteins consist of groups such as

the ULK–FIP200 complex, ATG9 vesicles, the class III phosphatidylinositol 3-kinase (PI3K) complex I, and the conjugation systems (ATG3, ATG5, ATG7, ATG10, ATG12, ATG16L1) that mediate LC3/GABARAP (ATG8) lipidation (Nakatogawa, 2020; Yamamoto et al, 2023). Autophagy was once thought to be responsible only for nonselective bulk degradation, but it is now also known to selectively degrade substrates. Studies have demonstrated that entire organelles such as mitochondria, endoplasmic reticulum (ER), lysosomes, and peroxisomes are selectively targeted into autophagosomes for elimination to maintain their quality or abundance in response to starvation or other stresses (Anding and Baehrecke, 2017; Gatica et al, 2018; Johansen and Lamark, 2020). These types of selective autophagy are called mitophagy, ER-phagy, lysophagy, pexophagy, and so on. Recently, the Golgi apparatus was also reported to be a substrate for autophagy ("Golgiphagy") (Lu et al, 2020; Nthiga et al, 2021; Rahman et al, 2022), but this has been less studied than the corresponding processes for other organelles.

The organelles to be degraded by selective autophagy are recognized directly or indirectly by ATG8 family proteins (LC3/GABARAP family proteins) on forming autophagosomes (Johansen and Lamark, 2020; Mizushima, 2020). In some cases, the cargo/organelles include proteins that directly interact with ATG8 family proteins and function as autophagy receptors. In other cases, autophagy adaptors such as SQSTM1/p62, NBR1, NDP52, TAX1BP1, and OPTN mediate the interaction between the cargo and ATG8 proteins, which often depends on the ubiquitination of the cargo. Most of the autophagy receptors or adaptors have a sequence called the LC3-interacting region (LIR) motif or GABARAP-interacting motif (GIM) (Birgisdottir et al, 2013; Rogov et al, 2017; Stolz et al, 2014). The canonical core LIR motif consists of $[W/F/Y]_0$-$X_1$-$X_2$-$[L/V/I]_3$ (X is any amino acid), in which conserved positions $X_0$ and $X_3$ are aromatic and hydrophobic residues, respectively (Johansen and Lamark, 2020).

In ER-phagy, six transmembrane ER-resident proteins (FAM134A/B/C, RTN3L, SEC62, ATL3, CCPG1, TEX264) have been reported as receptors (Chino and Mizushima, 2020), some of which contain reticulon homology domains (RHDs) that facilitate fragmentation of the ER membrane to meet the demands for ER-

[1]Department of Genetics, Graduate School of Medicine, Osaka University, Suita, Osaka 565-0871, Japan. [2]Integrated Frontier Research for Medical Science Division, Institute for Open and Transdisciplinary Research Initiatives (OTRI), Osaka University, Suita, Osaka 565-0871, Japan. [3]Laboratory of Intracellular Membrane Dynamics, Graduate School of Frontier Biosciences, Osaka University, Suita, Osaka 565-0871, Japan. [4]Division of Cell Signaling, Fujii Memorial Institute of Medical Sciences, Institute of Advanced Medical Sciences, Tokushima University, Tokushima 770-8503, Japan. [5]Faculty of Life Sciences, Kyoto Sangyo University, Motoyama, Kamigamo, Kita, Kyoto 603-8555, Japan. [6]Department of Biochemistry and Molecular Biology, Graduate School of Medicine, The University of Tokyo, Bunkyo-ku, Tokyo 113-0033, Japan. ✉E-mail: tamyoshi@fbs.osaka-u.ac.jp; kuma@fbs.osaka-u.ac.jp

phagy (Gubas and Dikic, 2022). Meanwhile, no organelle-resident autophagy receptors for Golgiphagy have been reported in mammals. It is reasonable to assume that each organelle has its own resident proteins that act as receptors for selective autophagy, which prompted us to search for Golgiphagy receptors.

In this study, we performed a proteomic analysis of *Atg5*-deficient mouse tissues, and found that Golgi-resident transmembrane proteins YIPF3 and YIPF4 accumulated in the autophagy-deficient tissues. We developed two Golgiphagy-reporter systems and demonstrated that the YIPF3–YIPF4 complex functions as a Golgiphagy receptor. YIPF3 contains the core LIR motif and putative phosphorylation sites upstream of the LIR, both of which are required for the interaction with ATG8 family proteins. YIPF4 is also required for Golgiphagy, at least for stabilizing the YIPF3–YIPF4 complex. Interestingly, the sequence of the phosphorylation sites is exactly the same as that of TEX264, the major ER-phagy receptor, implying that a similar regulatory mechanism might exit.

## Results

### Quantitative proteomic analysis identifies the Golgi transmembrane proteins YIPF3 and YIPF4 as candidates for autophagy cargo

In selective autophagy, autophagy receptors are incorporated into autophagosomes through interaction with the cargo and ATG8s, and degraded together with them in lysosomes. Correspondingly, many kinds of autophagy receptors have been shown to accumulate in autophagy-deficient cells and tissues. To identify novel autophagy receptors, we performed quantitative proteomic analysis of *Atg5*-deficient mouse brain by using multiplexed isobaric tandem mass tag (TMT) labeling. As expected, autophagy adaptors (p62/SQSTM1, NBR1, TAX1BP1) and ER-phagy receptors (FAM134a, FAM134c, CCPG1, RTN3, SEC62, CALCOCO1) clearly accumulated in *Atg5*-deficient tissues (Fig. 1A; Dataset EV1), confirming that our proteomic analysis was working. Among the Golgi-resident proteins, YIPF3 and YIPF4 were the most increased in *Atg5*-deficient tissues. YIPF3 and YIPF4 are *cis*-Golgi-localized proteins that have five transmembrane segments, with their N- and C-terminal regions exposed to the cytosol and Golgi lumen, respectively (Shaik et al, 2019) (Fig. 1B). Using AlphaFold-Multimer (preprint: Evans et al, 2022; Jumper et al, 2021), both proteins were predicted to have additional helices and unfolded regions mainly at their N-terminus, and also to make intimate contacts with each other through their transmembrane segments (Fig. 1B). Indeed, we observed that YIPF3 and YIPF4 form a complex, with the latter being required for the former's stability, as previously reported (Tanimoto et al, 2011) (Figs. 1C and EV1A). It was suggested that the YIPF3–YIPF4 complex is involved in maintaining the Golgi structure, but little is known about its functions. In this study, we focused on YIPF3 and YIPF4 because the Golgi membrane proteins involved in autophagy are not well known.

### YIPF3–YIPF4-positive Golgi fragments are delivered to lysosomes via autophagy

First, we observed the intracellular localization of YIPF3 and YIPF4 during autophagy induction. We generated mouse embryonic fibroblasts (MEFs) stably expressing YIPF3 or YIPF4 tagged with EGFP at the cytosolic N-terminus. Both EGFP–YIPF3 and EGFP–YIPF4 showed a juxtanuclear ribbon-like Golgi structure, and large proportions of EGFP–YIPF3 and EGFP–YIPF4 were colocalized with the *cis*-Golgi marker GM130 (Fig. EV2A), suggesting that both proteins were present in the Golgi. The Golgi apparatus has been suggested to be fragmented during starvation (Lu et al, 2020; Nthiga et al, 2021; Takahashi et al, 2011). We did not observe the fragmentation of Golgi in MEFs during starvation; however, with bafilomycin A$_1$ (Baf A$_1$), a vacuolar ATPase inhibitor, we observed the significant appearance of punctate structures of EGFP–YIPF3 and EGFP–YIPF4 in the cytoplasm, in addition to the ribbon-like Golgi structure (Fig. 2A,B). This suggested that YIPF3- and YIPF4-positive Golgi fragments were formed in response to starvation, and delivered to lysosomes. We further observed that many of the EGFP–YIPF3 and EGFP–YIPF4 puncta colocalized with the autophagosome marker LC3 under starvation conditions in the presence of Baf A$_1$ (EGFP–YIPF3, $63.8 \pm 8.1\%$; EGFP–YIPF4, $62.5 \pm 10.8\%$) (Fig. 2C). They also partially colocalized with or were in close proximity to the lysosomal marker LAMP1 (Fig. EV2B). These findings suggest that YIPF3 and YIPF4 were delivered to lysosomes via autophagy.

To confirm that the puncta of EGFP–YIPF3 and EGFP–YIPF4 were fragments of the Golgi apparatus, we observed EGFP–YIPF4 using several Golgi markers: GM130 (*cis*-Golgi), MAN2A1 (*medial*-Golgi), and TMEM165 (*trans*-Golgi). We first confirmed that EGFP–YIPF4 colocalized with endogenous YIPF3, suggesting that EGFP–YIPF4 adequately represents the YIPF3–YIPF4 complex (Fig. 2D). In untreated cells, GM130, TMEM165, and MAN2A1–mCherry were colocalized with EGFP–YIPF4 as juxta-nuclear ribbon-like Golgi structures (Fig. EV2A,C). Upon starvation with Baf A$_1$, these Golgi proteins formed punctate structures and colocalized with EGFP–YIPF4 (GM130, $41.2 \pm 2.7\%$; MAN2A1–mCherry, $63.3 \pm 6.8\%$; TMEM165, $59.0 \pm 9.0\%$) and LC3, suggesting that the EGFP–YIPF4 puncta represent the Golgi fragments targeted by autophagosomes (Fig. 2E). We further confirmed that endogenous YIPF3 formed punctate structures and colocalized with GM130 (*cis*-Golgi) and GRASP55 (*medial*-Golgi) under both growing and starvation conditions with Baf A$_1$ (GM130, $42.8 \pm 5.4\%$; GRASP55, $68.3 \pm 7.9\%$) (Fig. EV2D). Taken together, these findings suggest that the YIPF3- and YIPF4-positive fragments are formed from the Golgi apparatus and delivered to lysosomes via autophagy in response to starvation.

### YIPF3 and YIPF4 are degraded by autophagy

To clarify whether the YIPF3–YIPF4 complex is degraded by autophagy, we measured the protein levels of endogenous YIPF3 and YIPF4 during starvation. YIPF3 has three forms: I, 40 kDa; II, 46 kDa; and III, 36 kDa. YIPF3 is N-glycosylated in the ER (form I) and trafficked to the Golgi to be O-glycosylated (form II) and cleaved at its luminal region in the C-terminal (form III) (Tanimoto et al, 2011). Upon starvation, the amounts of YIPF3 (forms II and III) and YIPF4 were found to be reduced in a time-dependent manner, and these reductions were abolished in FIP200-KO cells (Fig. 3A,B). These results suggested that endogenous YIPF3 and YIPF4 were preferably degraded by autophagy. Meanwhile, CALCOCO1 showed more drastic degradation than YIPF3 and YIPF4 during starvation. Because CALCOCO1 was reported to be an ER-phagy and Golgiphagy

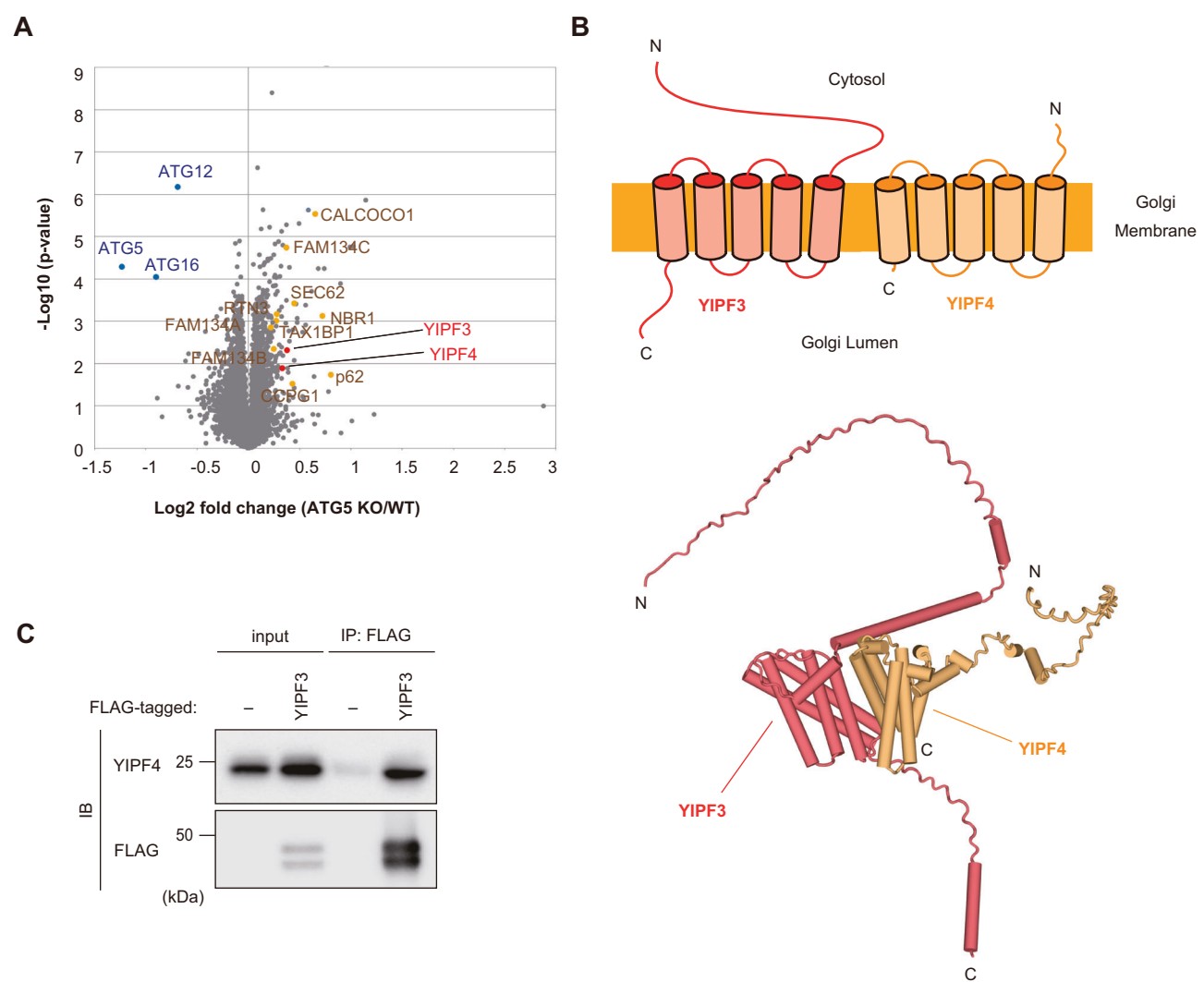

**Figure 1. The Golgi transmembrane proteins, YIPF3 and YIPF4, accumulated in autophagy-deficient mouse tissues.**

(A) A volcano plot of the TMT-based quantitative proteomics showing changes in the proteins of $Atg5^{+/+}$ and $Atg5^{-/-}$ mouse brain homogenates. A paired analysis of variance together with a Student's $t$ test was used for statistical processing of the results. (B) Schematic of YIPF3 and YIPF4 topology. Both proteins are suggested to have five transmembrane segments, with their N- and C-terminal regions exposed to the cytoplasm and Golgi lumen, respectively. Structures of YIPF3 (deep salmon) and YIPF4 (light orange) predicted by AlphaFold-Multimer are shown as cartoons. (C) Immunoprecipitation of FLAG–YIPF3 in HEK293T cells. Cells were transfected with FLAG–YIPF3. The lysates were immunoprecipitated with anti-FLAG antibodies and detected with anti-YIPF4 and anti-FLAG antibodies. Source data are available online for this figure.

receptor/co-receptor (Nthiga et al, 2020; Nthiga et al, 2021), the results might reflect that CALCOCO1 was degraded via ER-phagy and Golgiphagy upon starvation. Considering that GM130 and TMEM165 were colocalized with YIPF4 upon starvation (Fig. 2E), we measured the levels of these Golgi proteins. Both GM130 and TMEM165 were accumulated in FIP200-KO cells (Fig. 3A,B) consistently with a previous report (Nthiga et al, 2021), suggesting the Golgi apparatus was degraded via autophagy.

We further demonstrated that YIPF3 and YIPF4 were degraded by autophagy using Halo tag. Processing assays using Halo tag have recently been reported (Yim et al, 2022). The ligand-bound Halo tag becomes resistant to lysosomal proteolysis, and thus the amount of free Halo processed from Halo–mGFP–YIPF3 or Halo–mGFP–YIPF4 by lysosomal proteases reflects their degradation in lysosomes delivered

via autophagy (Fig. 3C) (Yim et al, 2022). The cleavage of a free Halo tag from Halo–mGFP–YIPF3 or Halo–mGFP–YIPF4 was increased by starvation, which was completely abolished by Baf $A_1$ treatment, confirming that the free Halo tag was generated in a lysosome-dependent manner (Fig. 3D,F). The increased processing of Halo–mGFP–YIPF3 or Halo–mGFP–YIPF4 upon starvation was suppressed by the deletion of FIP200, showing that YIPF3 and YIPF4 were degraded by autophagy upon starvation (Fig. 3D–G).

## YIPF3 interacts with specific ATG8s through cytosolic N-terminal LIR motif

Interaction with ATG8s is crucial for autophagy receptors to incorporate the cytoplasmic cargo into autophagosomes. Because

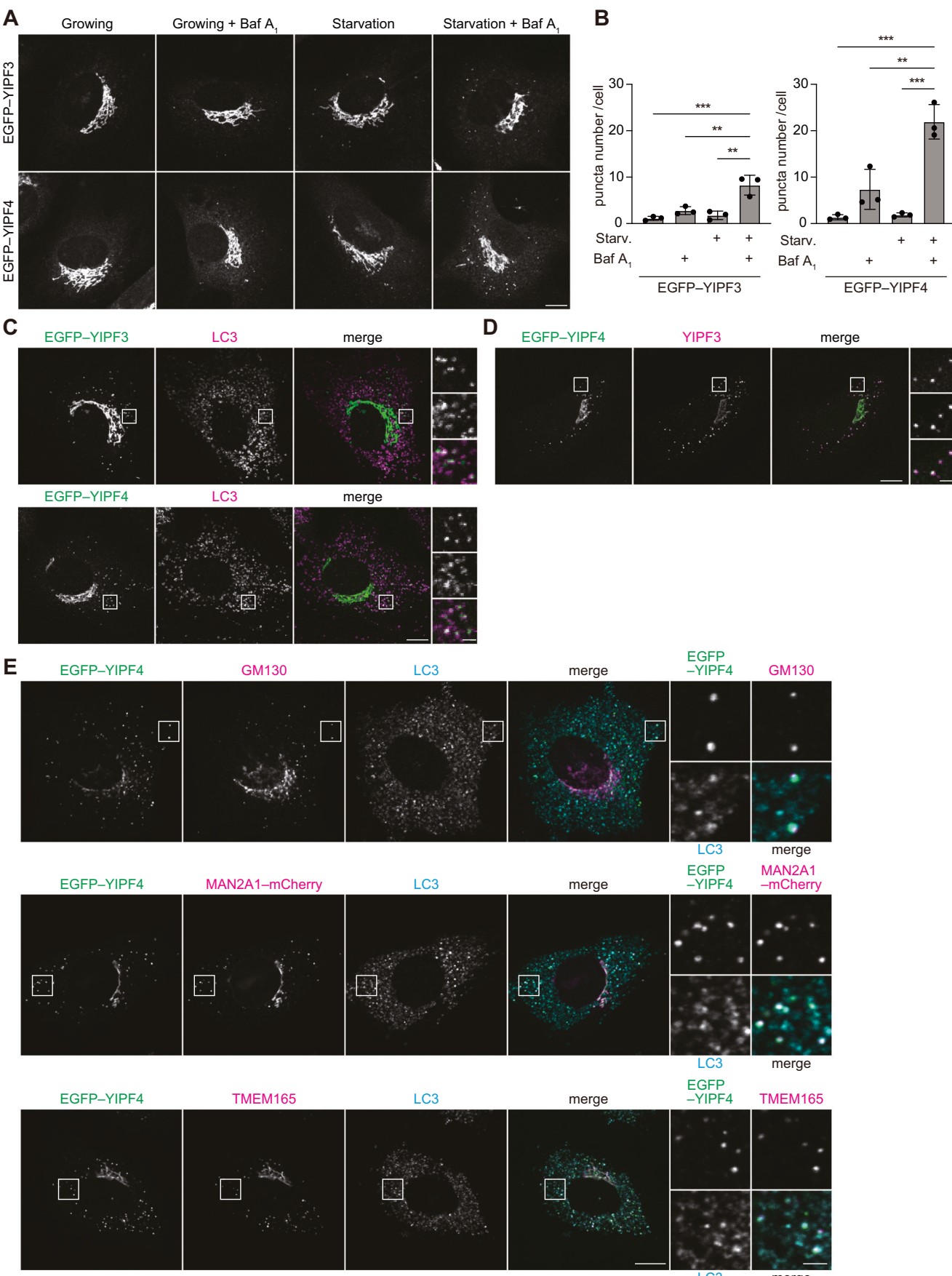

**Figure 2. YIPF3- and YIPF4-positive Golgi fragments are trafficked to autophagosomes.**

(A) Immunofluorescence images of MEFs stably expressing EGFP–YIPF3 or EGFP–YIPF4. Cells were cultured in DMEM or EBSS (4 h) in the presence or absence of 125 nM Baf A₁. Scale bars, 10 μm. (B) Quantification of the number of EGFP–YIPF3 or EGFP–YIPF4 puncta per cell in the cytosol in cells shown in (A). More than 60 cells were analyzed per condition in each experiment. Error bars indicate mean ± SD ($n = 3$ independent experiments). (C) Immunofluorescence image of MEFs stably expressing EGFP–YIPF3 or EGFP–YIPF4. Cells were cultured in EBSS (2 h) in the presence of 125 nM Baf A₁ and stained with anti-LC3 antibodies. Scale bars, 10 μm and 1 μm (insets). The ratio of EGFP puncta positive for LC3 to total EGFP puncta was quantified in three independent experiments (mean ± SD). Ten cells were analyzed per condition in each experiment. (D) Immunofluorescence image of MEFs stably expressing EGFP–YIPF4. Cells were cultured in EBSS (6 h) in the presence of 125 nM Baf A₁ and stained with anti-YIPF3 antibodies. Scale bars, 10 μm and 1 μm (insets). (E) Immunofluorescence image of MEFs stably expressing EGFP–YIPF4 or coexpressing EGFP–YIPF4 and MAN2A1–mCherry. Cells were cultured in EBSS (6 h) in the presence of 125 nM Baf A₁. Each cell was stained with LC3, and the indicated cells were stained with anti-GM130 or anti-TMEM165 antibodies. Scale bars, 10 μm and 2 μm (insets). The ratio of EGFP–YIPF4 puncta positive for each Golgi protein to total EGFP–YIPF4 puncta was quantified in three independent experiments (mean ± SD). Ten cells were analyzed per condition in each experiment. Data information: $P$ values were determined using one-way ANOVA with Tukey's multiple comparisons test (B). Symbols indicate: **$P \leq 0.01$, ***$P \leq 0.001$. Source data are available online for this figure.

YIPF3 contains three evolutionarily conserved putative LIR motifs in the N-terminal region that face the cytosol (Fig. 4A), we examined whether YIPF3 interacts with ATG8s. Coimmunoprecipitation analysis revealed that endogenous YIPF3 interacted with FLAG-tagged LC3B, GABARAP, and GABARAPL1, whereas YIPF4 did not show clear interaction with any ATG8s (Fig. 4B). Similarly, FLAG–YIPF4 did not bind to endogenous LC3 or GABARAPL1, whereas they were clearly shown to interact with FLAG–YIPF3 (Fig. 4C).

Next, we tested whether YIPF3 interacts with ATG8s via LIR motifs. We introduced mutations into three putative LIR motifs: YIPF3LIR2A_1 (F47A, M50A), YIPF3LIR2A_2 (Y112A, I115A), and YIPF3LIR2 A_3 (Y121Z, V124 A) (Fig. 4A). These mutations did not affect the binding to YIPF4 (Fig. 4D). Of the three, only the YIPF3LIR2A_1(F47A, M50A) mutant lost the binding to LC3 and GABARAPL1, showing that amino acid residues 47–50 constitute the LIR motif required for the binding of YIPF3 to ATG8s (Fig. 4D). We observed that the LIR mutant correctly localized on the Golgi apparatus when stably expressed in MEFs; however, it caused morphological changes such as partially discontiguous and expanded Golgi structures under basal conditions (Fig. EV3A,B), indicating the importance of Golgi turnover via selective autophagy.

## mRFP–EGFP–Golgi reporter reveals that the YIPF3–YIPF4 complex is required for Golgiphagy

The YIPF3–YIPF4 complex binds to ATG8 family proteins via the LIR motif, similarly to other autophagy receptors, and therefore we investigated whether the YIPF3–YIPF4 complex functions as a receptor for Golgiphagy. To this end, we established two assays for measuring Golgiphagic activity. First, we modified the previous assays using an mRFP–EGFP tandem fluorescent tag, which is often used to measure autophagy flux. mRFP–EGFP-tagged proteins such as mRFP–EGFP–LC3 show both mRFP and EGFP signals before fusing with lysosomes; however, mRFP-only signals (mRFP⁺, EGFP⁻) appear when they are delivered to lysosomes because EGFP has lower stability relative to mRFP in the acidic environment there (Katayama et al, 2008). Thus, the appearance of mRFP-only puncta (mRFP⁺, EGFP⁻) represents autophagic activity. In previous studies, a tandem fluorescent tag was fused to short sequences of mitochondrial proteins FIS or COXVIII for mitophagy (Allen et al, 2013; Zhu et al, 2011), and the ER-retention signal KEDL for ER-phagy (Chino et al, 2019); therefore, we used a minimal Golgi-targeting motif FLWRIFCFRK for analyzing

Golgiphagy (Navarro and Cheeseman, 2022). This Golgi-targeting motif has recently been reported and validated as a Golgi marker (Navarro and Cheeseman, 2022). We placed mRFP–EGFP–FLWRIFCFRK (hereinafter referred to as mRFP–EGFP–Golgi) under the control of a doxycycline-inducible promoter to suppress background signals of mRFP (Fig. 5A). To confirm the feasibility of this system, we first observed the localization of mRFP–EGFP–Golgi. HeLa cells stably expressing mRFP–EGFP–Golgi were treated with doxycycline for 24 h. After washing out the reagents, cells were cultured for an additional 9 h under growing or starvation conditions. Under growing conditions, mRFP–EGFP–Golgi was colocalized with GM130 (*cis*-Golgi) and p230 (*trans*-Golgi) (Fig. 5B), showing that mRFP–EGFP–Golgi was properly targeted to the Golgi apparatus, as previously reported (Navarro and Cheeseman, 2022). After 9 h of starvation, the mRFP-only puncta (mRFP⁺, EGFP⁻) appeared, which were abolished in FIP200-KO cells (Fig. 5C,D). This suggested that the mRFP-only puncta (mRFP⁺, EGFP⁻) represented the Golgi fragments delivered into lysosomes via autophagy. Thus, we concluded that the mRFP–EGFP–Golgi reporter can be used to measure Golgiphagy flux visually by microscopy.

Using this system, we examined whether YIPF3 and YIPF4 were required for Golgiphagy. We introduced the mRFP–EGFP–Golgi reporter to YIPF3-KO or YIPF4-KO HeLa cells generated by the CRISPR–Cas9 system. We observed that the deletion of YIPF4 suppressed the YIPF3 levels, resulting in the loss of both proteins, as previously reported (Tanimoto et al, 2011) (Fig. EV1A). No clear changes in cell growth or the morphology of the Golgi apparatus were observed upon the deletion of YIPF3 and YIPF4 (Fig. EV1B). Under both growing and starvation conditions, the appearance of mRFP-only puncta (mRFP⁺, EGFP⁻) was partially suppressed in YIPF3-KO and YIPF4-KO cells compared with that in WT cells (Fig. 5C,D). These findings suggest that YIPF3 and YIPF4 are required for Golgiphagy.

## Halo–mGFP–Golgi reporter-processing assay reveals that the YIPF3–YIFP4 complex is required for Golgiphagy

Next, we developed a Halo tag-based processing assay for Golgiphagy (Fig. 6A). Halo tag can be detected by in-gel fluorescence, which is more quantitative than western blotting (Yim et al, 2022). Taking advantage of the Halo tag's features, we developed the Halo–mGFP–FLWRIFCFRK (hereinafter referred to as Halo–mGFP–Golgi) reporter to measure Golgiphagic activity quantitatively. We validated the system by measuring autophagy

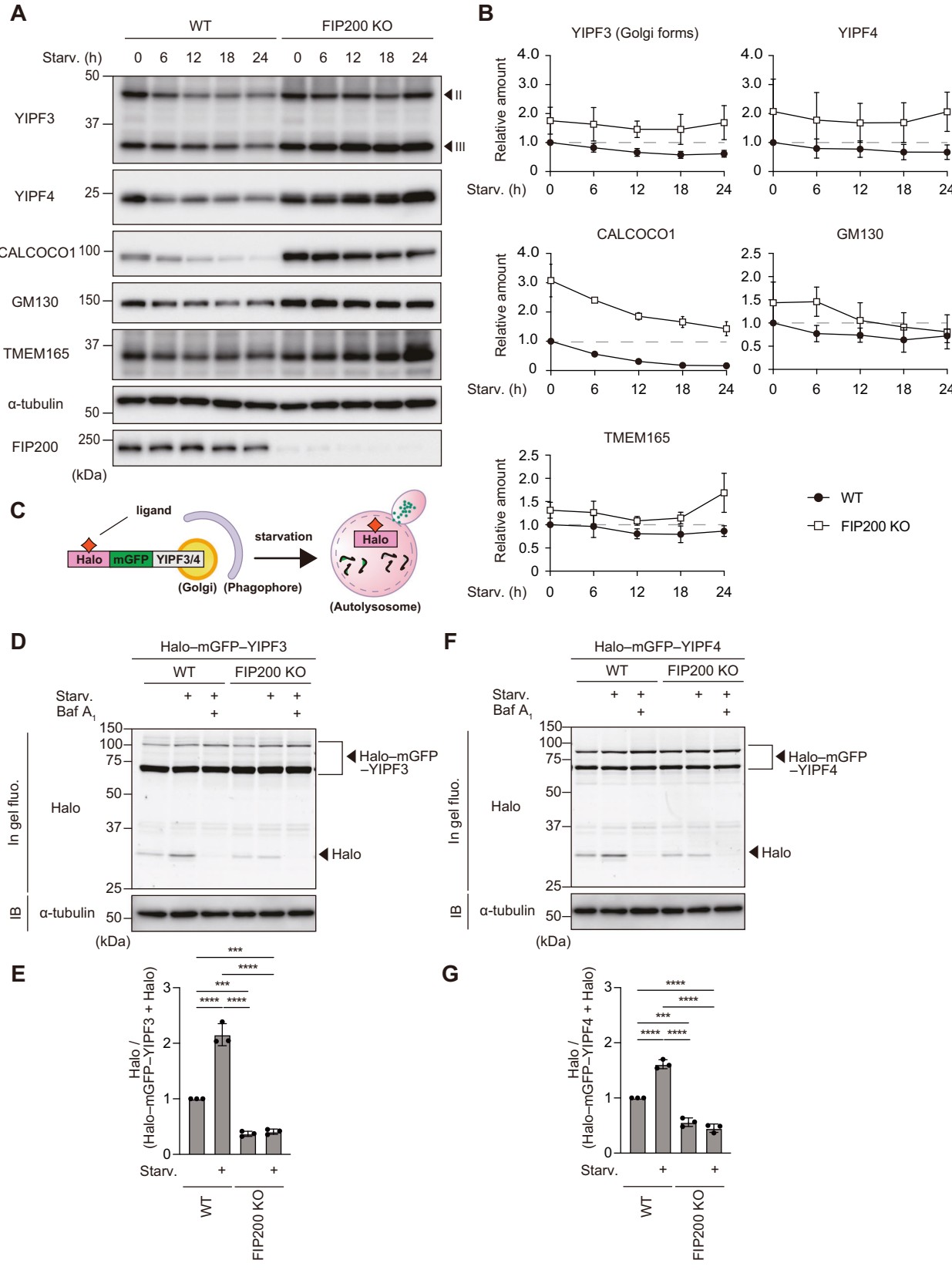

**Figure 3. YIPF3 and YIPF4 are autophagic substrates.**

(A) Immunoblotting of the indicated proteins in WT and FIP200-KO HeLa cells under nutrient-rich conditions or starvation conditions for 6, 12, 18, and 24 h. (B) Quantification of results shown in (A). Each band intensity of the indicated proteins is normalized with that of α-tubulin. Relative amounts normalized to WT cultured in the nutrient-rich medium are shown. Error bars indicate mean ± SD ($n = 3$ independent experiments). (C) Schematic of processing assay using Halo–mGFP–YIPF3 or Halo–mGFP–YIPF4. The GFP signal is quenched by the acidic pH and the linker between mGFP and Halo tag is cleaved by lysosomal proteases in the autolysosomes, while Halo with ligand signal remains stable. (D) In-gel fluorescence image of Halo in WT and FIP200-KO HeLa cells stably expressing Halo–mGFP–YIPF3. After cells were cultured in the presence of TMR-conjugated ligands for 20 min and washed out, they were cultured in DMEM or EBSS in the presence or absence of 125 nM Baf $A_1$ for 9 h. (E) Quantification of results shown in (D). The ratio of Halo:Halo–mGFP–YIPF3 is shown. Each ratio is normalized to WT cultured in nutrient-rich medium. Error bars indicate mean ± SD ($n = 3$ independent experiments). (F) In-gel fluorescence image of Halo in WT and FIP200-KO HeLa cells stably expressing Halo–mGFP–YIPF4. After cells were cultured in the presence of TMR-conjugated ligands for 20 min and washed out, they were cultured in DMEM or EBSS in the presence or absence of 125 nM Baf $A_1$ for 9 h. (G) Quantification of results shown in (F). The ratio of Halo:Halo–mGFP–YIPF4 is shown. Each ratio was normalized to WT cultured in nutrient-rich medium. Error bars indicate mean ± SD ($n = 3$ independent experiments). Data information: P values were determined using one-way ANOVA with Tukey's multiple comparisons test (E, G). Symbols indicate: ***$P \leq 0.001$, ****$P \leq 0.0001$. Source data are available online for this figure.

flux and comparing it between WT and FIP200-KO cells, or upon treatment with or without Baf $A_1$. HeLa cells stably expressing Halo–mGFP–Golgi were treated with tetramethylrhodamine (TMR)-conjugated Halo ligands for 20 min. After washing out the ligands, the cells were cultured for an additional 9 h under nutrient-rich or starvation conditions. Generation of the free Halo tag (33 kDa) by the cleavage of Halo–mGFP–Golgi was increased upon starvation in WT cells, while this was abolished in FIP200-KO cells or by Baf $A_1$ treatment (Fig. 6B,C). This confirmed that the appearance of free Halo tag was dependent on autophagy and lysosomal degradation. These findings showed that the Halo–mGFP–Golgi reporter can be used to measure Golgiphagy flux quantitatively by SDS-PAGE.

We then examined whether YIPF3 and YIPF4 were involved in Golgiphagy. The amount of free Halo tag was significantly lower in YIPF3-KO cells and YIPF4-KO cells than in WT cells under growing or starvation conditions (Fig. 6B,C). Consistent with the results of microscopic analysis shown in Fig. 5, the Golgiphagic activity was suppressed in YIPF3-KO cells to an extent similar to that seen in YIPF4-KO cells, suggesting that the loss of only YIPF3 is sufficient to suppress Golgiphagy, and YIPF4 is required for Golgiphagy by stabilizing YIPF3 (Fig. 6B,C). Taking these findings together, we concluded that the YIPF3–YIPF4 complex is required for Golgiphagy. We also examined whether YIPF3 and YIPF4 were involved in nonselective bulk autophagy and ER-phagy using Halo–LC3 and Halo–mGFP–KDEL reporters, respectively (Yim et al, 2022). Lack of YIPF3 and YIPF4 affected neither nonselective autophagy nor ER-phagy (Fig. EV4A–D). YIPF3 and YIPF4 did not colocalize with FIP200 puncta during starvation (Fig. EV4E), implying that YIPF3 and YIPF4 were not present at autophagosome formation sites, unlike ATG9 vesicles from the Golgi. These results suggest that YIPF3 and YIPF4 are specifically involved in Golgiphagy.

The deletion of YIPF3 and YIPF4 clearly suppressed Golgiphagy, but not completely. This indicates the existence of other Golgiphagy receptors in addition to the YIPF3–YIPF4 complex. Indeed, CALCOCO1 was recently reported to be a soluble Golgiphagy receptor/co-receptor (Nthiga et al, 2021). Thus, we compared the contribution of these Golgiphagy receptors to starvation-induced Golgiphagy using our Halo–mGFP–Golgi assay. HeLa cells stably expressing Halo–mGFP–Golgi were treated with each combination of siRNAs against YIPF3, YIPF4, CALCOCO1, and FIP200. The Golgiphagic activity was suppressed by about 30% under growing conditions and about 20% under starvation conditions in YIPF3/4 double-knockdown cells compared with

the level in control cells, whereas in CALCOCO1-knockdown cells it was comparable to the level in control cells (Fig. 6D,E). Knockdown of both YIPF3 and YIPF4 with the additional knockdown of CALCOCO1 had no additive effect on Golgiphagy (Fig. 6D,E). These findings indicated at least that CALCOCO1 functions in a pathway distinct from that in which the YIPF3–YIPF4 complex functions.

## Overexpression of YIPF3–YIPF4 complex upregulates Golgiphagic activity

Some ER-phagy receptors such as FAM134B, RTN3L, and TEX264 have been reported to enhance ER-phagy when overexpressed (An et al, 2019; Chino et al, 2019; Grumati et al, 2017; Khaminets et al, 2015), and therefore we investigated whether the overexpression of YIPF3 and YIPF4 is sufficient to enhance Golgiphagy. HA–YIPF3 and/or FALG–YIPF4 were stably expressed in HeLa cells harboring Halo–mGFP–Golgi reporter. The morphology of the Golgi was not affected by overexpression of YIPF3 and YIPF4 (Appendix Fig. S1). However, we noticed that the expression levels of YIPF3 appeared to depend on the amount of YIPF4. It has been reported that endogenous YIPF3 was decreased and replaced by exogenous YIPF3 when it was overexpressed because YIPF3 required YIPF4 for its stability (Tanimoto et al, 2011). Consistent with this, we observed that only when HA–YIPF3 and FLAG–YIPF4 were expressed together did the total level of both proteins (sum of endogenous and exogenous proteins) successfully increase. Meanwhile, single expression of HA–YIPF3 or FLAG–YIPF4 resulted in a decreased level of endogenous YIPF3 or YIPF4, and failed to increase the total amount of YIPF3 or YIPF4 (Fig. 7A,B). Correspondingly, Golgiphagy flux was upregulated in cells coexpressing both HA–YIPF3 and FLAG–YIPF4, but not in the cells overexpressing only one of these under both growing and starvation conditions (Fig. 7A,C). These findings suggest that overexpression of the YIPF3–YIPF4 complex is sufficient to induce Golgiphagy, and also imply that the formation of a complex between YIPF3 and YIPF4 is required for them to function in Golgiphagy in HeLa cells.

## YIPF3–YIPF4 complex functions as a Golgiphagy receptor

To clarify whether Golgiphagy is mediated by the YIPF3 LIR-ATG8 interaction, we performed a Halo–mGFP–Golgi processing assay in YIPF3-KO cells re-expressing HA–YIPF3 or HA–YIPF3 LIR2A_1 mutant. The amount of free Halo tag was

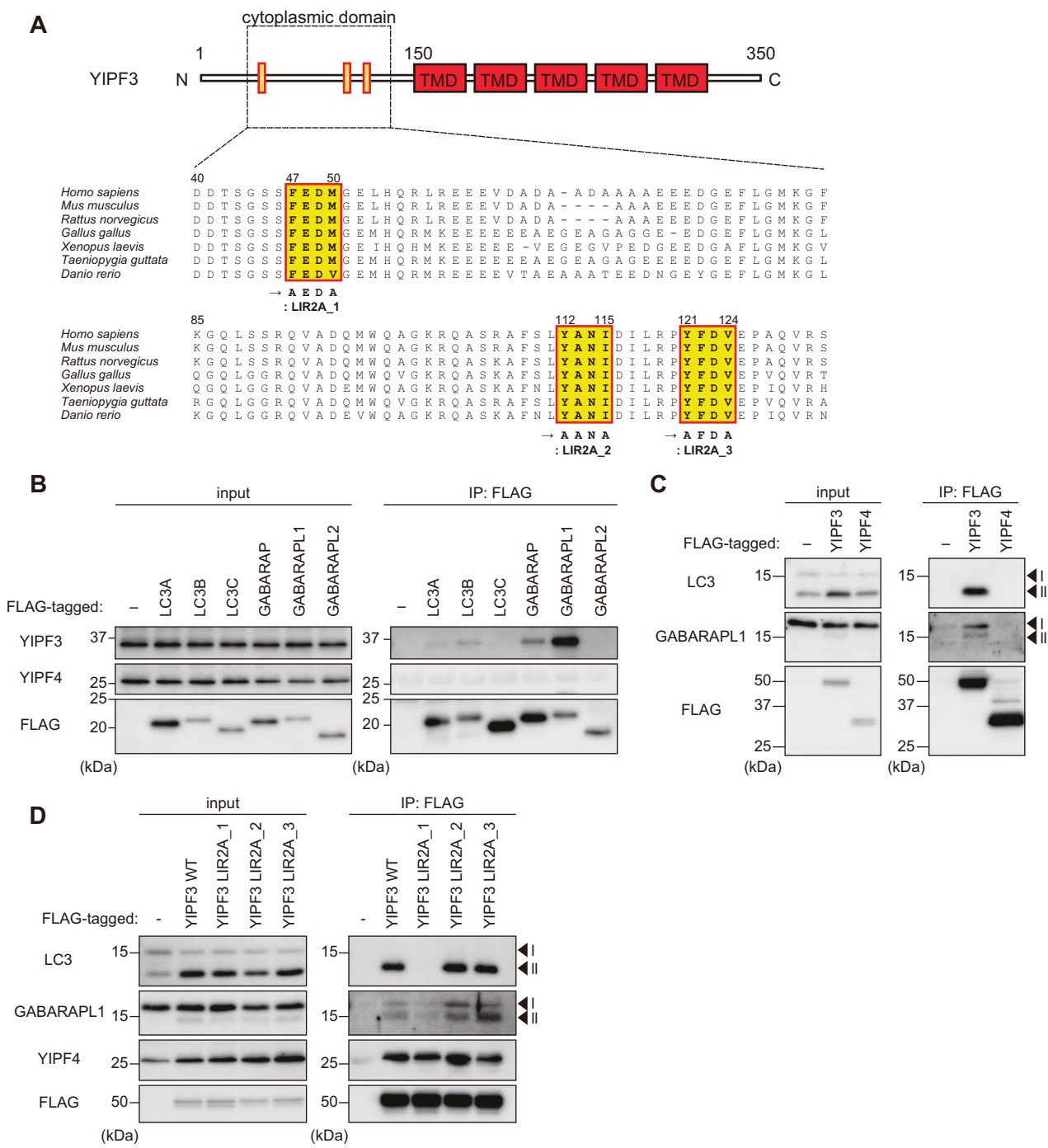

**Figure 4. YIPF3 binds to specific ATG8s via cytosolic N-terminal LIR motif.**

(A) Schematic of YIPF3 domain structure and the alignment of N-terminal regions containing evolutionarily conserved putative LIR motifs in vertebrates. TMD, transmembrane domain. (B) Immunoprecipitation of FLAG–ATG8s in HEK293T cells. Cells were transfected with FLAG–ATG8s. The lysates were immunoprecipitated with anti-FLAG antibody and detected with anti-YIPF3, anti-YIPF4, and anti-FLAG antibodies. (C) Immunoprecipitation of FLAG–YIPF3 and FLAG–YIPF4 in HEK293T cells. Cells were transfected with FLAG–YIPF3 or FLAG–YIPF4. The lysates were immunoprecipitated with anti-FLAG antibody and detected with anti-LC3, anti-GABARAPL1, and anti-FLAG antibodies. (D) Immunoprecipitation of FLAG–YIPF3 WT or LIR2A mutants in HEK293T cells. Cells were transfected with FLAG–YIPF3 WT or each LIR2A mutant. The lysates were immunoprecipitated with anti-FLAG antibody and detected with anti-LC3, anti-GABARAPL1, anti-YIPF4, and anti-FLAG antibodies. Source data are available online for this figure.

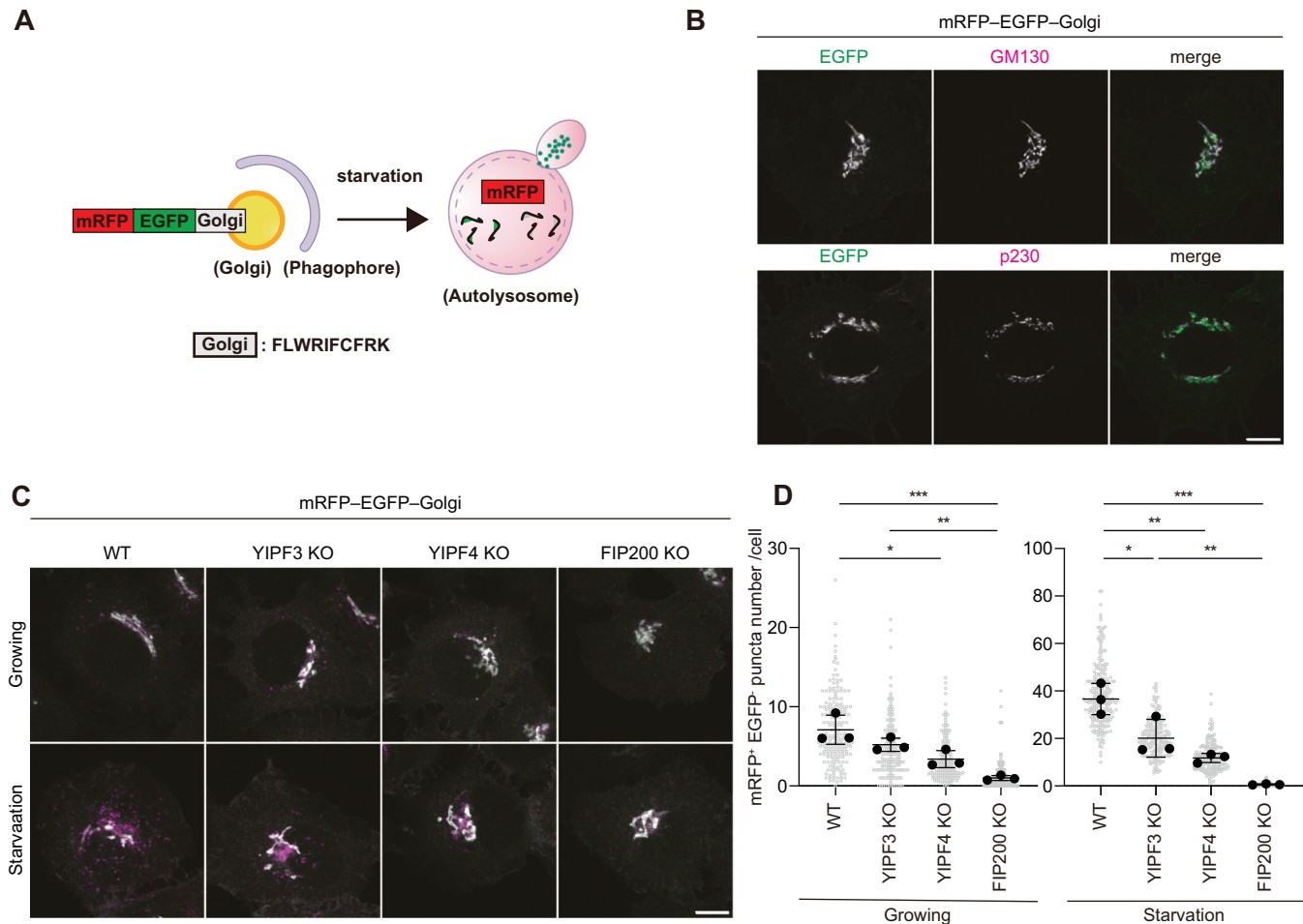

**Figure 5. mRFP–EGFP–Golgi reporter reveals that YIPF3–YIPF4 complex mediates Golgiphagy.**

(A) Schematic of microscopy analysis using mRFP–EGFP–Golgi as a reporter for Golgiphagy. Golgi consists of a Golgi-targeting motif FLWRIFCFRK. mRFP is stable, but EGFP is quenched and degraded in autolysosomes. mRFP-positive but EGFP-negative puncta indicate Golgi fragments delivered into lysosomes. (B) Immunofluorescence image of HeLa cells stably expressing mRFP–EGFP–Golgi under growing conditions. Cells were stained with anti-GM130 or -p230 antibodies. Scale bars, 10 μm.
(C) Immunofluorescence image of WT, YIPF3-KO, YIPF4-KO, and FIP200-KO HeLa cells stably expressing mRFP–EGFP–Golgi. After cells were cultured in the presence of doxycycline for 24 h and washed out, they were cultured in DMEM or EBSS for 9 h. Scale bars, 10 μm. (D) Quantification of the number of mRFP$^+$, EGFP$^-$ puncta from mRFP–EGFP–Golgi per cell in WT, YIPF3-KO, YIPF4-KO, and FIP200-KO HeLa cells stably expressing mRFP–EGFP–Golgi. More than 100 cells were analyzed per condition in each experiment. Error bars indicate mean ± SD ($n = 3$ independent experiments). Data information: $P$ values were determined using one-way ANOVA with Tukey's multiple comparisons test (D). Symbols indicate: $*P \leq 0.05$; $**P \leq 0.01$, $***P \leq 0.001$. Source data are available online for this figure.

decreased by the deletion of YIPF3 compared with that in WT cells, which was restored by re-expressing HA–YIPF3 to the same level as in WT cells (Fig. 8A,B). Meanwhile, HA–YIPF3 LIR2A_1 mutant did not recover the Golgiphagic activity (Fig. 8A,B), showing that Golgiphagy is dependent on the YIPF3 LIR motif. In addition, more YIPF4 accumulated in YIPF3-KO cells expressing YIPF3 LIR2A_1 mutant than in WT and YIPF3-KO cells expressing HA–YIPF3 (Fig. 8A), indicating that the degradation of YIPF4 is dependent on the YIPF3 LIR motif. These findings suggest that the YIPF3–YIPF4 complex functions as a Golgiphagy receptor.

## YIPF3–YIPF4-mediated Golgiphagy depends on the phosphorylation of YIPF3 LIR

Binding of the autophagy receptors to the ATG8 family proteins via the LIR motif is crucial in selective autophagy. The upstream sequence of LIRs often contains negatively charged residues such as Asp, Glu, and phosphorylated Ser and Thr, which enhance the binding with ATG8s by forming electrostatic interactions in addition to the canonical interaction within the LIR motif (Noda et al, 2010).

Interestingly, YIPF3 contains the same upstream sequence ($S_{-4}$-$G_{-3}$-$S_{-2}$-$S_{-1}$) as that of the ER-phagy receptor TEX264, in which phosphorylation of the serine residues at positions −2 and −1 is critical for its interaction with the ATG8 proteins by forming multiple hydrogen bonds (Chino et al, 2022) (Fig. 9A). Indeed, phosphorylation of YIPF3 was detected by anti-phosphoserine antibody (Fig. 9B). This prompted us to generate structural models of the YIPF3–YIPF4 heterodimer bound with the ATG8 proteins using AlphaFold-Multimer (preprint: Evans et al, 2022; Jumper et al, 2021) to gain molecular insight into their interaction. Plausible complex interfaces were predicted only by using the

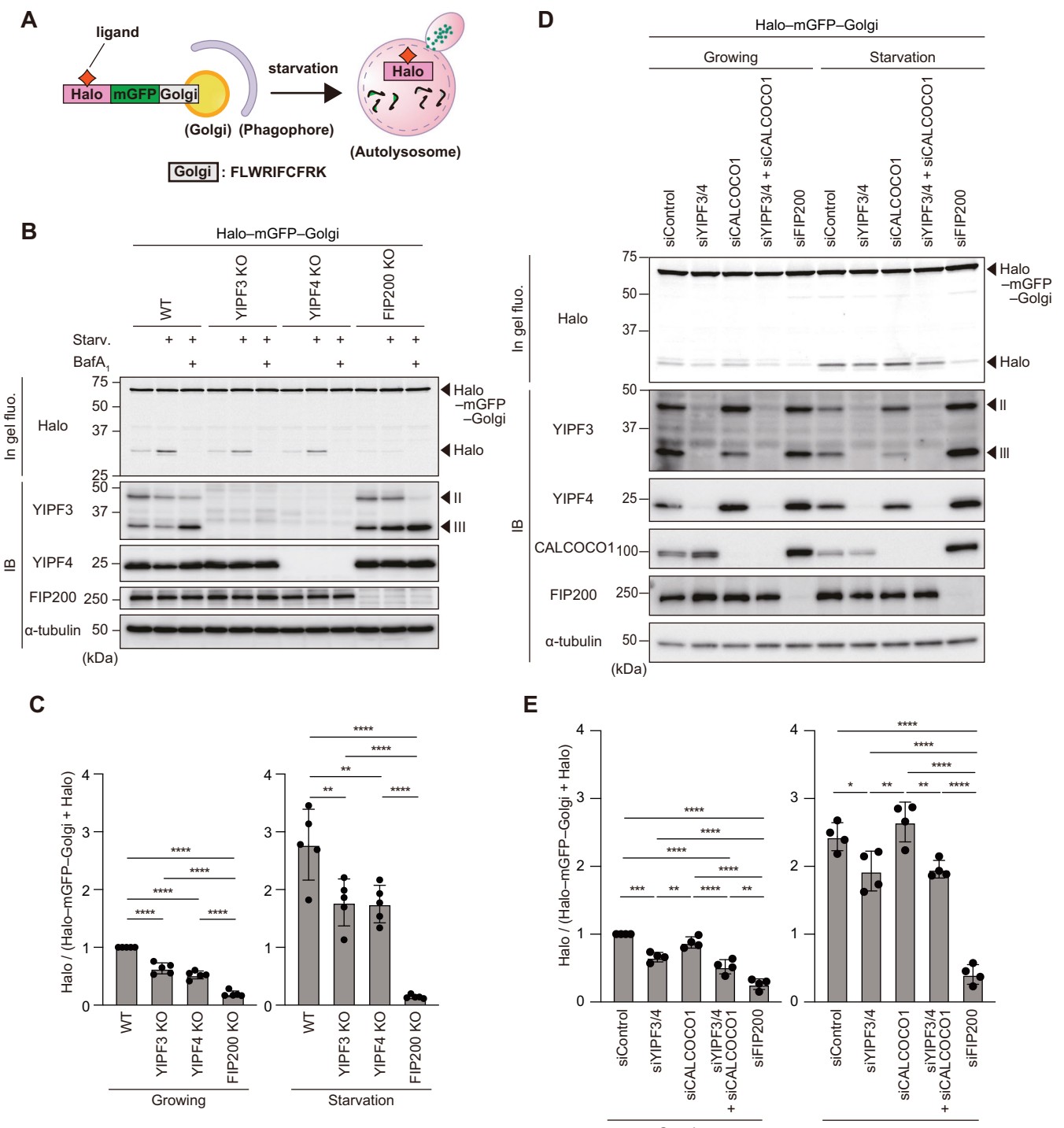

YIPF3 sequence with both Ser45 and Ser46 mutated to Asp or Glu, and the acidic residues were further replaced by phosphoserines in a representative of the calculated structures. In the resulting model of the complex (Fig. 9C, left), the LIR motif at the N-terminus of YIPF3 was found in an extended conformation to dock with the β-sheet of GABARAPL1, with the side chains of Phe47 and Met50 inserted into hydrophobic pockets, consistent with the typical mode of binding the LIR motif to the ATG8 protein (Fig. 9C, left).

In addition, the phosphoserine residues were located within hydrogen-bonding distances of His9, Arg47, and Lys48, which is reminiscent of the possible hydrogen bonds formed between those of the TEX264 LIR and GABARAP (Fig. 9C, right). This further indicates the importance of the phosphorylation of the serine residues next to the LIR motif for the interaction. To investigate whether the corresponding serine residues at positions $X_{-1}$ and $X_{-2}$ in the YIPF3 LIR are important for the interaction with the

**Figure 6. Processing assay using Halo–mGFP–Golgi reporter reveals that YIPF3–YIFP4 complex regulates Golgiphagy.**

(A) Schematic of processing assay using Halo–mGFP–Golgi as a reporter for Golgiphagy. Golgi consists of a Golgi-targeting motif FLWRIFCFRK. Halo tag with the ligand is stable, but GFP is degraded in autolysosomes. The amount of free Halo tag indicates cleavage of the reporter through autophagy. (B) In-gel fluorescence image of Halo and immunoblotting of indicated proteins in WT, YIPF3-KO, YIPF4-KO, and FIP200-KO HeLa cells stably expressing Halo–mGFP–Golgi. After cells were cultured in the presence of TMR-conjugated ligands for 20 min and washed out, they were cultured in DMEM or EBSS in the presence or absence of 125 nM Baf $A_1$ for 9 h. (C) Quantification of results is shown in (B). The ratio of Halo:Halo–mGFP–Golgi is shown. Each ratio is normalized to WT cultured in DMEM. Error bars indicate mean ± SD ($n = 5$ independent experiments). (D) In-gel fluorescence image of Halo and immunoblotting of indicated proteins in HeLa cells stably expressing Halo–mGFP–Golgi under treatment with siRNAs against the combination of YIPF3/4, CALCOCO1, combination of YIPF3/4 and CALCOCO1, FIP200, or control. After siRNA treatment for 72 h, cells were cultured in the presence of TMR-conjugated ligands for 20 min and washed out, and then cultured in DMEM or EBSS for 9 h. (E) Quantification of results shown in (D). The ratio of Halo:Halo–mGFP–Golgi is shown. Each ratio is normalized to WT cultured in DMEM. Error bars indicate mean ± SD ($n = 4$ independent experiments). Data information: $P$ values were determined using one-way ANOVA with Tukey's multiple comparisons test (C, E). Symbols indicate: *$P \le 0.05$; **$P \le 0.01$, ***$P \le 0.001$, ****$P \le 0.0001$. Source data are available online for this figure.

ATG8s, we replaced Ser45 and Ser46 with Ala (Fig. 9A). Phosphorylation of YIPF3 detected by anti-phosphoserine antibody was diminished in the YIPF3$^{S45A, S46A}$ and YIPF3$^{S43A, S45A, S46A}$ mutants (Fig. 9B). The YIPF3$^{S45A, S46A}$ mutant lost the interaction with LC3 and GABARAPL1 (Fig. 9D), suggesting that the phosphorylation of YIPF3 is required for its interaction with ATG8s. Meanwhile, the phosphomimic YIPF3$^{S45D, S46D}$ and YIPF3$^{S45E, S46E}$ mutants showed stronger affinity to LC3 compared with the YIPF3$^{S45A, S46A}$ mutant, but less affinity than YIPF3 WT (Fig. EV5). This is analogous to the interaction of TEX264 with GABARAP that was barely restored by Asp or Glu mutations of the serine residues at the corresponding positions (Chino et al, 2022).

Finally, to examine whether the phosphorylation of Ser45 and Ser46 of YIPF3 is required for Golgiphagy, we performed the Halo–mGFP–Golgi processing assay in YIPF3-KO cells re-expressing HA–YIPF3 or HA–YIPF3$^{S45A, S46A}$. HA–YIPF3 recovered Golgiphagic activity, but HA–YIPF3 $^{S45A, S46A}$ did not (Fig. 9E,F), suggesting that the phosphorylation of YIPF3 at Ser45 and Ser46 is required for Golgiphagy by strengthening its binding to ATG8s via the LIR motif. Taking these findings together, we proposed that the YIPF3–YIPF4 complex is the cargo-resident autophagy receptor for Golgiphagy, which directly interacts with ATG8s, probably in a fashion similar to that for TEX264 (Fig. 9G).

## Discussion

In this study, we identified the YIPF3–YIPF4 complex as a selective autophagy receptor for degradation of the Golgi apparatus, which is the first Golgi-resident membrane protein to be identified as an autophagy receptor in mammals. The interaction of this complex with LC3B, GABARAP, and GABARAPL1 is dependent on YIPF3's LIR motif and putative phosphorylation sites immediately upstream, while its stability is governed by YIPF4. In addition, overexpression of the YIPF3 LIR mutant led to an expansion of the Golgi apparatus, indicating the importance of Golgi turnover via selective autophagy. Golgiphagy is less well studied than other organelle-specific types of autophagy such as mitophagy and ER-phagy. So far, CALCOCO1 and GMAP have been reported as Golgiphagy receptors/co-receptors in mammals and fly, respectively (Nthiga et al, 2021; Rahman et al, 2022). GOLPH3 is also a putative Golgiphagy receptor in mammals (Lu et al, 2020). CALCOCO1 is a soluble receptor/co-receptor for both Golgiphagy and ER-phagy, which requires the interaction with cargo-anchoring proteins ZDHHC17/13 on the Golgi for Golgiphagy or VAPA/B on

the ER for ER-phagy (Nthiga et al, 2020; Nthiga et al, 2021). Meanwhile, YIPF3 and YIPF4 are the first Golgi membrane proteins known to function as a Golgiphagy receptor. During the review of our manuscript, Harper's group published a related paper, in which they also identified YIPF3 and YIPF4 as Golgiphagy receptors in elegant analyses using mass spectrometry (Hickey et al, 2023).

In mammals, multiple ER-phagy receptors such as FAM134A/B/C, RTN3L, SEC62, ATL3, TEX264, CCGP1, and CALCOCO1 have been identified (Chino and Mizushima, 2020; Nthiga et al, 2020). Similarly, multiple receptors may be involved in Golgiphagy because YIPF3, YIPF4, and CALCOCO1 triple-knockdown cells still retain some Golgiphagic activity. We observed that the deletion of YIPF3 or YIPF4 resulted in more than 35% decrease in Golgiphagy flux, as judged by our Halo–mGFP–Golgi reporter assay (Fig. 6C), indicating that the YIPF3–YIPF4 complex is responsible for not a small amount of the Golgiphagy among the multiple receptors that may exist. Unexpectedly, knockdown of CALCOCO1 did not decrease Golgiphagy, judging from our Halo–mGFP–Golgi processing assay (Fig. 6D,E). Harper's group also reported that proteomic analysis showed that CALCOCO1-KO cells exhibited an extent of Golgi membrane protein turnover comparable to that of control cells (Hickey et al, 2023). These observations indicate that the YIPF3–YIPF4 complex may contribute to Golgiphagy more than CALCOCO1.

YIPF3 contains the core LIR motif and putative phosphorylation sites upstream of the LIR, both of which are required for the interaction with ATG8 family proteins. Although we did not show direct evidence for the phosphorylation of Ser45 and Ser46 of YIPF3, the sequence of the phosphorylation sites is exactly the same as that of TEX264, the major ER-phagy receptor, implying that a similar regulatory mechanism might exit. Because Ser271 and Ser272 of TEX264 are phosphorylated by casein kinase2 (Chino et al, 2022), YIPF3 may also be phosphorylated by this kinase. Although YIPF3 and YIPF4 were reported to exist manly at the *cis*-Golgi (Tanimoto et al, 2011), they were well colocalized with GM130 (*cis*-Golgi), MAN2A1–mCherry, (*medial*-Golgi), GRASP55 (*medial*-Golgi) and TMEM165 (*trans*-Golgi) on the Golgi fragments during starvation (Figs. 2E and EV2D). This indicates that YIPF3–YIPF4-mediated Golgiphagy can target all subcompartments of the Golgi.

Few assays have been established to evaluate Golgiphagic activity. In previous studies, Golgiphagic activity was estimated based on the colocalization of Golgi marker proteins with LC3 or the degradation of Golgi proteins as measured by western blotting (Lu et al, 2020; Rahman et al, 2022). Recently, tandem fluorescent-based markers

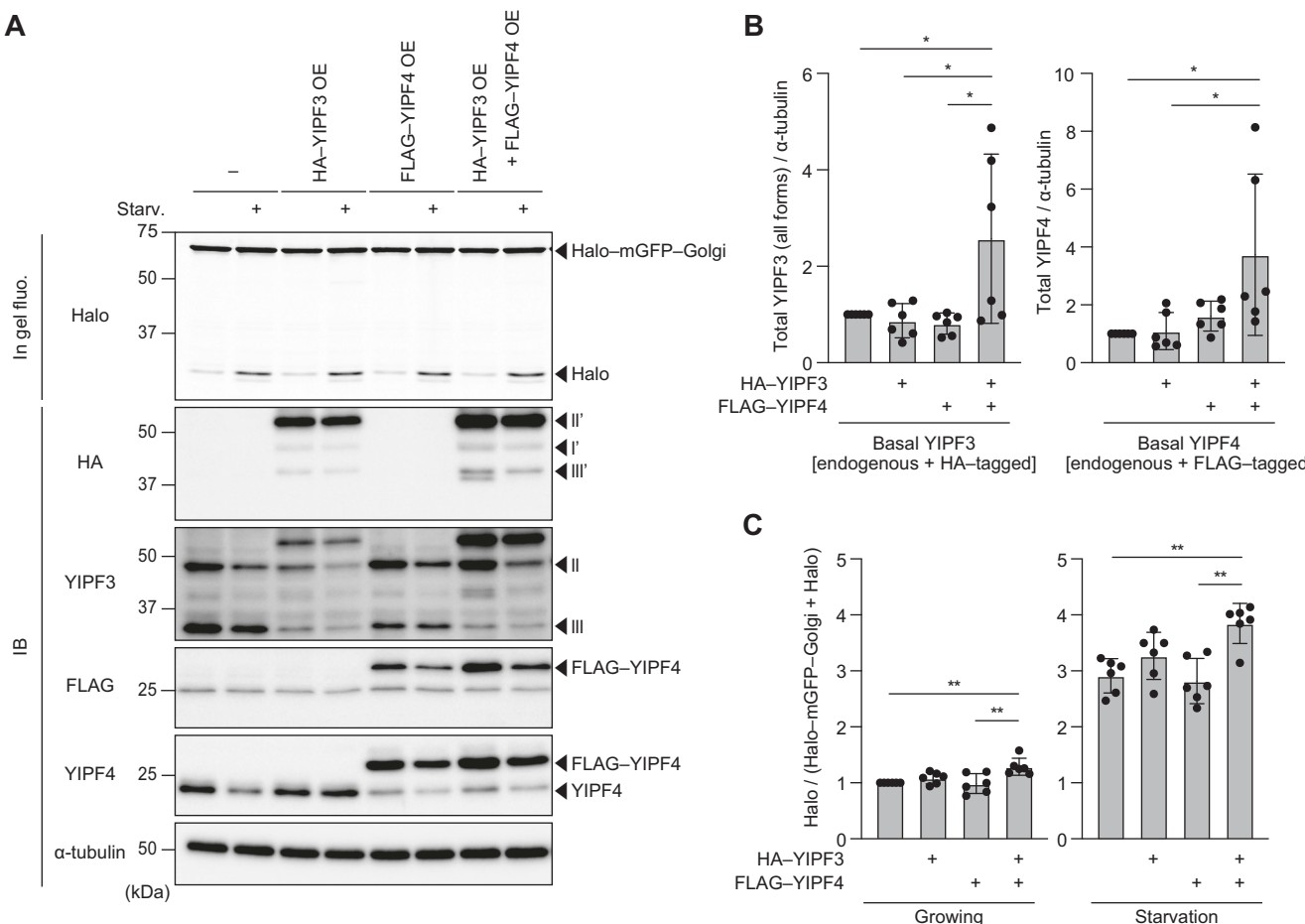

**Figure 7. Overexpression of YIPF3–YIPF4 upregulates Golgiphagic flux.**

(A) In-gel fluorescence image of Halo and immunoblotting of indicated proteins in HeLa cells stably expressing Halo-mGFP-Golgi alone or coexpressing HA-YIPF3 and/or FLAG-YIPF4 together. After cells were cultured in the presence of TMR-conjugated ligands for 20 min and washed out, they were cultured in DMEM or EBSS for 9 h. (B) Quantification of the intensity of the total (both endogenous and exogenous) YIPF3 (all forms) or YIPF4 bands in (A). Error bars indicate mean ± SD ($n = 6$ independent experiments). (C) Quantification of results of processing assay by Halo-mGFP-Golgi shown in (A). The ratio of Halo:Halo-mGFP-Golgi is shown. Each ratio is normalized to Control (cells stably expressing only Halo-mGFP-Golgi) cultured in nutrient-rich medium. Error bars indicate mean ± SD ($n = 6$). Data information: $P$ values were determined using one-way ANOVA with Tukey's multiple comparisons test (B, C). Symbols indicate: $*P \leq 0.05$; $**P \leq 0.01$. Source data are available online for this figure.

mCherry–EYFP–ZDHHC17 and mCherry–EYFP–TMEM165 were also used to analyze Golgiphagy (Nthiga et al, 2021). However, we observed that ZDHHC17 and TMEM165 localized to other structures in addition to the Golgi apparatus, which is consistent with previously reported findings (Rosnoblet et al, 2013; Singaraja et al, 2002). Therefore, we decided to develop a novel reporter for monitoring Golgiphagy. To monitor selective autophagy, a rather short sequence such as FIS[101–152] or the presequence of COXVIII (mitophagy), KDEL (ER-phagy), and SKL (pexophagy) are used to target probes to the cargo (Allen et al, 2013; Chino et al, 2019; Marcassa et al, 2018; Zhu et al, 2011). Thus, we tested a 10-amino-acid sequence that has been shown to be necessary and sufficient for Golgi localization (Navarro and Cheeseman, 2022). The mRFP–EGFP–Golgi reporter was at least partially colocalized with GM130 (*cis*-Golgi) and p230 (*trans*-Golgi) (Fig. 5B), even though the peptide was reported to be preferentially targeted to *cis*-Golgi. This indicates that this peptide might be suitable to monitor Golgiphagy regardless of the Golgi subcompartment. In addition to microscopic analysis, we developed the Halo tag-based

processing assay for Golgiphagy using Halo Tag Fluorescent Ligands because this is easier to perform and more quantitative for measuring Golgiphagy flux. These reporters, mRFP–EGFP–Golgi and Halo–mGFP–Golgi, should be helpful for further studies of Golgiphagy.

YIPF3 and YIPF4 are homologs of yeast Yif1 and Yip1, respectively. Yip1 and Yif1 have been shown to play essential roles in ER-to-Golgi transport in yeast; however, the function of YIPF proteins has not been well studied in mammals (Shaik et al, 2019). It was reported that the Golgi function was intact in YIPF3-KO or YIPF4-KO HeLa cells, even though morphological changes in the Golgi were observed (Tanimoto et al, 2011). Given the functions of Yip1 and Yif1 in yeast and the involvement of ATG9 vesicles from the Golgi in autophagosome formation (De Tito et al, 2020; Sawa-Makarska et al, 2020), there is a concern that the suppression of Golgiphagy in YIPF3-KO or YIPF4-KO cells might be due to the loss of function of the Golgi or membrane trafficking. However, we did not observe the morphological changes in the Golgi in our

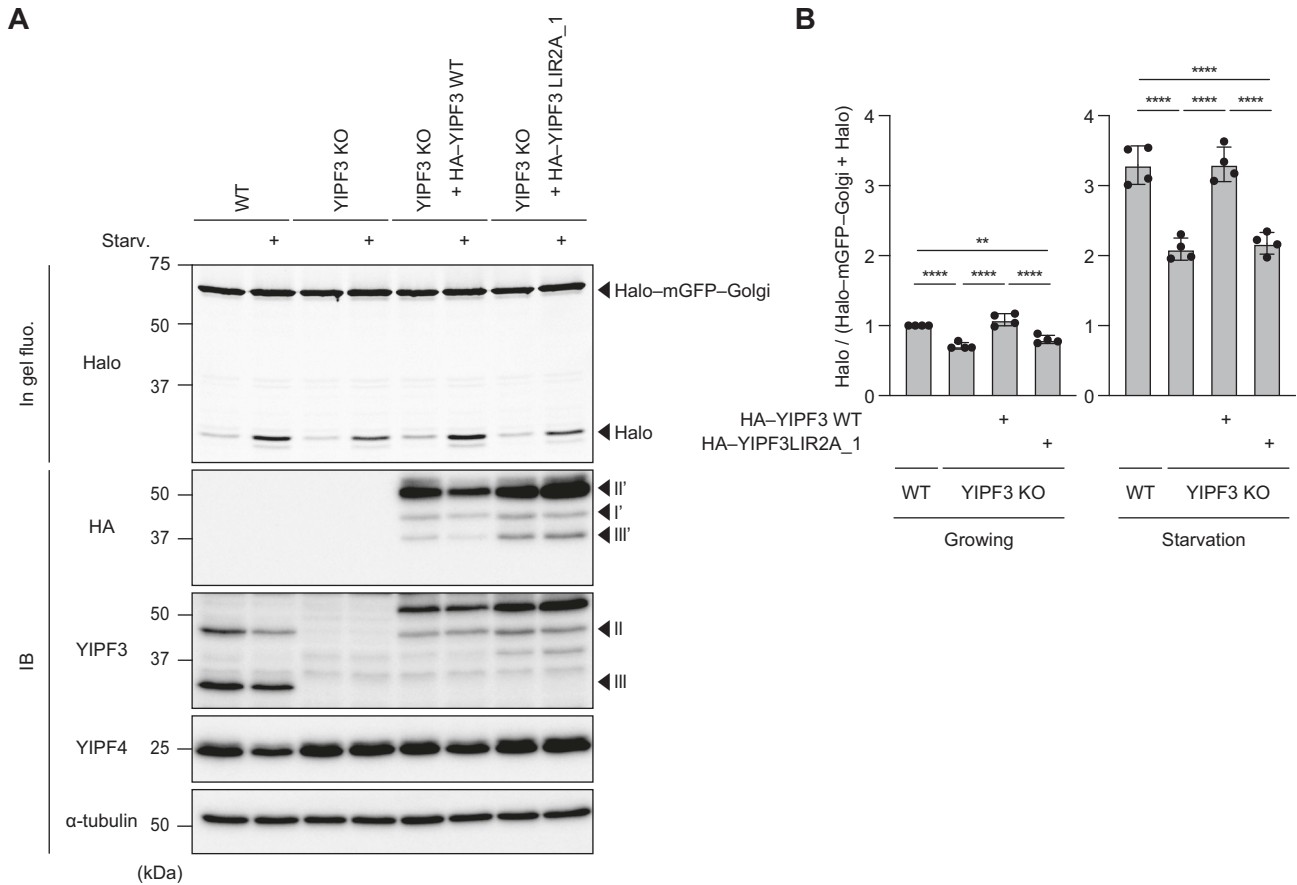

**Figure 8.  YIPF3 and YIPF4 are Golgiphagy receptors.**

(A) In-gel fluorescence image of Halo and immunoblotting of indicated proteins in WT and YIPF3-KO HeLa cells stably expressing Halo–mGFP–Golgi alone or coexpressing HA–YIPF3 or HA–YIPF3LIR2A_1 together. After cells were cultured in the presence of TMR-conjugated ligands for 20 min and washed out, they were cultured in DMEM or EBSS for 9 h. (B) Quantification of results shown in (A). The ratio of Halo:Halo–mGFP–Golgi is shown. Each ratio is normalized to WT cultured in nutrient-rich medium. Error bars indicate mean ± SD (n = 4 independent experiments). Data information: P values were determined using one-way ANOVA with Tukey's multiple comparisons test (B). Symbols indicate: **P ≤ 0.01, ****P ≤ 0.0001. Source data are available online for this figure.

HeLa cells (Fig. EV1B), and nonselective bulk autophagy and ER-phagy were normal in YIPF3-KO or YIPF4-KO cells (Fig. EV4). In addition, the replacement of just two amino acids in the LIR motif in YIPF3 was sufficient to suppress the Golgiphagic activity. Given these observations, the defects in Golgiphagy seen in YIPF3-KO or YIPF4-KO cells were more likely due to the YIPF3–YIPF4 complex losing its function as a receptor. We found that YIPF3 binds to ATG8 family proteins via the LIR motif, but YIPF4 did not show such interactions. We could not investigate whether YIPF4 exerts functions in Golgiphagy beyond stabilizing YIPF3 because deletion of YIPF4 led to the loss of YIPF3, which prevented us from identifying the effect of deleting YIPF4 in isolation. Instead, it would be interesting to find proteins that interact with YIPF4 during Golgiphagy to elucidate its function.

The physiological significance of Golgiphagy remains unclear. It has been reported that the deletion of CALCOCO1 caused expansion of the Golgi apparatus in mammalian cells and the LIR mutants of GMAP exhibited the accumulation of Golgi proteins and elongation of the Golgi in fly (Nthiga et al, 2021; Rahman et al, 2022). Very recently, Harper's group have shown that the YIPF3–YIPF4 complex mediates the recycling

of Golgi proteins during nutrient stress and neuronal differentiation in vitro (Hickey et al, 2023). Consistent with this, we observed that MEFs expressing the LIR mutant of YIPF3 showed morphological changes in the Golgi compared with WT MEFs (Fig. EV3). This was not observed in YIPF3-KO or YIPF4-KO HeLa cells, probably because some compensatory mechanism occurred. Taken together, these findings indicate the importance of Golgi turnover by selective autophagy.

Fragmented ER and mitochondria need to be engulfed by autophagosomes. Meanwhile, Golgi fragmentation is broadly observed in both physiological and pathological conditions, in which the Golgi ribbon structure becomes unlinked/unstacked, and is dispersed throughout the cytoplasm as tubules/vesicles (Chang and Yang, 2022; Wei and Seemann, 2017). Golgi fragmentation is induced during mitosis, under various Golgi stresses, and in pathological conditions such as apoptosis, infection, amyotrophic lateral sclerosis, Alzheimer's disease, Parkinson's disease, and cancer (Chang and Yang, 2022; Wei and Seemann, 2017), in addition to starvation (Lu et al, 2020; Nthiga et al, 2021; Takahashi et al, 2011). At present, Golgiphagy has only been observed upon starvation. It would be interesting to determine whether

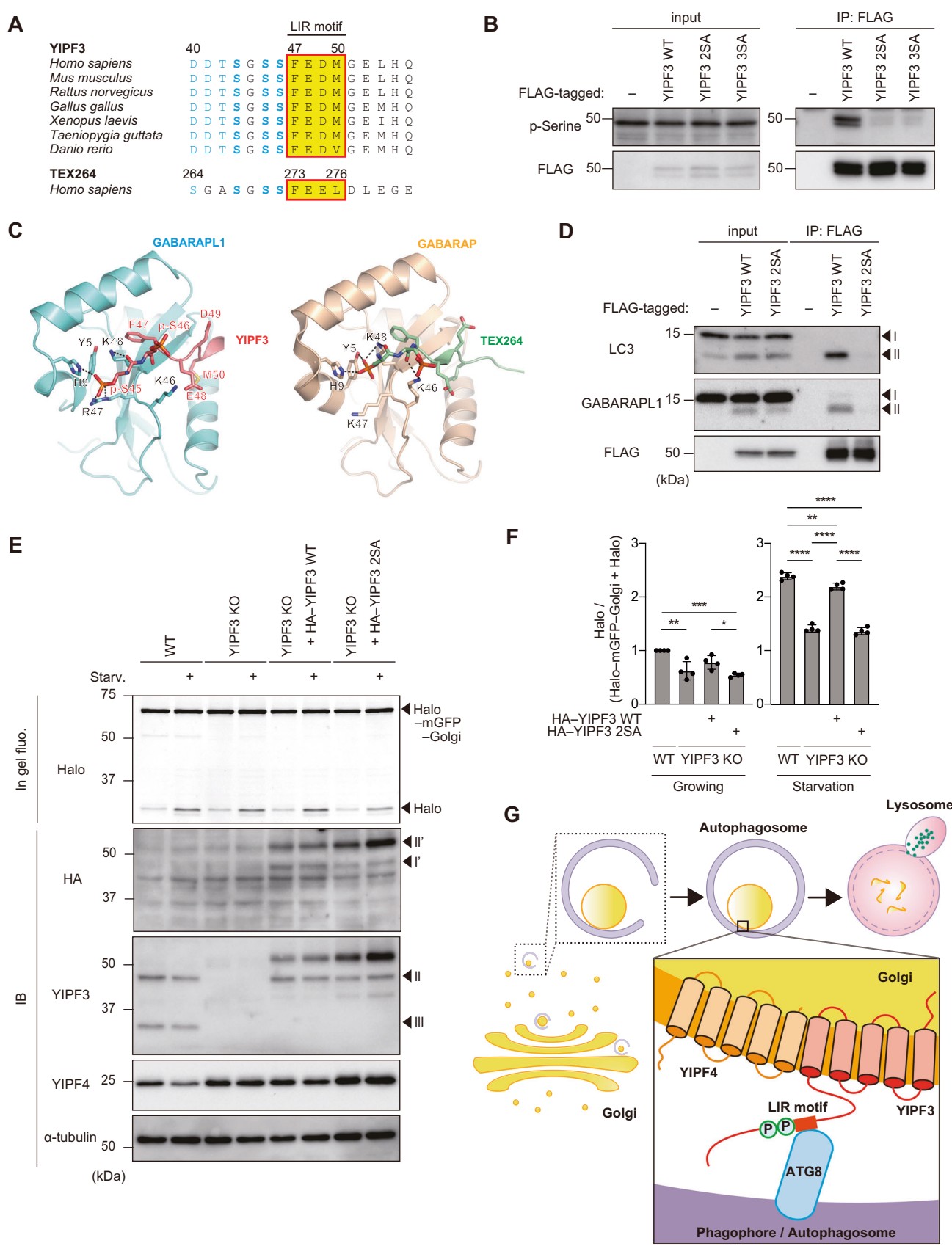

**Figure 9. YIPF3–YIPF4-mediated Golgiphagy depends on the phosphorylation of serine residues upstream of the LIR motif on YIPF3.**

(A) Alignment of YIPF3 LIR and its surroundings in vertebrates and TEX264 LIR in human showing the existence of negatively charged residues (shown in blue). Ser43, Ser45 and Ser46 are mutation sites for alanine substitution (shown in bold blue). (B) Immunoprecipitation of FLAG–YIPF3, FLAG–YIPF3$^{S45A, S46A}$ (2SA) or FLAG–YIPF3$^{S43A, S45A, S46A}$ (3SA) in HEK293T cells. Cells were transfected with FLAG–YIPF3 or each serine mutants. The lysates were immunoprecipitated with anti-FLAG antibody and detected with anti-phosphoserine and anti-FLAG antibodies. (C) A structural model of the YIPF3 LIR (deep salmon) bound to GABARAPL1 (light orange) based on predictions by AlphaFold-Multimer in comparison with the crystal structure of the TEX264 LIR (pale green) bound to GABARAP (wheat) (PDB accession code: 7VEC). Hydrogen bonds that can potentially form between the LIRs and the GABARAP proteins are shown by black dotted lines. (D) Immunoprecipitation of FLAG–YIPF3 or YIPF3$^{S45A, S46A}$ (2SA) in HEK293T cells. Cells were transfected with FLAG–YIPF3 or YIPF3$^{S45A, S46A}$. The lysates were immunoprecipitated with anti-FLAG antibody and detected with anti-LC3, anti-GABARAPL1, and anti-FLAG antibodies. (E) In-gel fluorescence image of Halo and immunoblotting of indicated proteins in WT and YIPF3-KO HeLa cells stably expressing Halo-mGFP–Golgi alone or coexpressing HA-YIPF3 or HA-YIPF3$^{S45A, S46A}$ (2SA) together. After cells were cultured in the presence of TMR-conjugated ligands for 20 min and washed out, they were cultured in DMEM or EBSS for 9 h. (F) Quantification of results shown in (E). The ratio of Halo:Halo-mGFP–Golgi is shown. Each ratio is normalized to WT cultured in nutrient-rich medium. Error bars indicate mean ± SD ($n = 4$ independent experiments). (G) Model of how YIPF3–YIPF4-mediated Golgiphagy occurs and is regulated. Data information: $P$ values were determined using one-way ANOVA with Tukey's multiple comparisons test (F). Symbols indicate: *$P \le 0.05$; **$P \le 0.01$, ***$P \le 0.001$, ****$\le 0.0001$. Source data are available online for this figure.

Golgiphagy occurs under these stresses that induce Golgi fragmentation. Further studies should reveal the physiological relevance of Golgiphagy and its mechanisms, and our reporters should be useful for analyzing Golgiphagy and provide new insights into the quality control of the Golgi apparatus.

# Methods

## Antibodies and reagents

For immunoblotting, the following antibodies were used: anti-Halo (mouse, G9211; Promega), anti-α-tubulin (rabbit, PM054; MBL), anti-FLAG (goat, ab1257; Abcam), anti-HA (rat, 11867423001; Roche), anti-YIPF4 (rabbit, HPA017884; Atlas Antibodies), anti-GABARAPL1 (D5R9Y; Cell Signaling), anti-phosphoserine (ab9332; Abcam), anti-CALCOCO1 (mouse, sc-515670; Santa Cruz Biotechnology), anti-LC3 (rabbit, PM036; MBL), anti-TMEM165 (mouse, 15465-1-AP; Proteintech), anti-GM130 (mouse, 610822; BD Biosciences), and anti-FIP200 (rabbit, 17250-1-AP; Proteintech) antibodies. anti-GRASP55 and anti-YIPF3 antibodies were as previously described (Shorter et al, 1999; Tanimoto et al, 2011). Horseradish peroxidase (HRP)-conjugated goat anti-rabbit IgG (111-035-003; Jackson ImmunoResearch), HRP-conjugated goat anti-rat IgG (112-035-003; Jackson ImmunoResearch), HRP-conjugated goat anti-mouse IgG (115-035-003; Jackson ImmunoResearch), and HRP-conjugated rabbit anti-goat IgG (305-036-003; Jackson ImmunoResearch) were used as the secondary antibodies.

For immunofluorescence staining, the following antibodies were used: anti-LC3 (mouse, M152-3; MBL), anti-LAMP1 (rat, 553792; BD Biosciences), and anti-p230 (mouse, 611280; BD Biosciences) antibodies. Alexa Fluor 488-conjugated goat anti-rabbit IgG H&L (ab150085; Abcam), Alexa Fluor 568-conjugated goat anti-mouse IgG (H + L) (A11004; Invitrogen), Alexa Fluor 568-conjugated goat anti-rabbit IgG H&L (ab175695; Abcam), Alexa Fluor 647-conjugated goat anti-rabbit IgG (H + L) (A21245; Invitrogen), and Alexa Fluor 647-conjugated goat anti-mouse IgG H&L (ab150119; Abcam) were used as the secondary antibodies.

## Plasmids

pMRX–IB–Halotag7–mGFP–KDEL (#184904), pMRX–No–HaloTag7–mGFP–LC3–mRFP (#184902), and pDmyc-neo-N1-MAN2A1–GFP

(#163649) were obtained from Addgene. pENTR1A and pcDNA3.1 were purchased from Invitrogen. The pMRX–IRES–puro and pMRX–IRES–bsr vectors were provided by S. Yamaoka (Tokyo Medical and Dental University, Tokyo, Japan). pCW57–CMV–ssRFP–GFP–KDEL (#128257) was provided by N. Mizushima (Department of Biochemistry and Molecular Biology, The University of Tokyo). The plasmid ptfLC3 was as previously described (Kimura et al, 2007). The DNA fragments of human YIPF3 (NP_056203.2) and YIPF4 (NP_115688.1) were amplified from HeLa Kyoto cDNA. To generate pMRX–mCherry–MAN2A1, DNA fragments corresponding to MAN2A1 (1–116 aa) in pDmyc–neo-N1–MAN2A1–GFP were amplified. These fragments were subcloned into the EcoRI site (for YIPF3) or BamHI/XhoI site [for YIPF4 and MAN2A1 (1–116 aa)] of pENTR1A. pENTR1A adapters were transferred into pMRX–3xFLAG, pMRX–3xHA, pMRX–EGFP, pMRX–mCherry, and pcDNA3.1–3xFLAG using an LR (Gateway system) reaction (Invitrogen). For alanine substitutions of positions $X_1/X_4$ in YIPF3 LIR candidates or Ser45/Ser46 residues upstream of the LIR motif in YIPF3, PCR amplification was conducted using PrimeSTAR Max (Takara) or KOD-Plus-Neo (TOYOBO).

For the construction of the plasmid mRFP–EGFP–Golgi under the doxycycline-inducible promoter (Tet-On mRFP–EGFP Golgi), the Golgi sequence FLWRIFCFRK was inserted into the BglII/EcoRI site of ptfLC3, and mRFP–EGFP–Golgi fragments were amplified and inserted into the BamHI/EcoRI site of pCW57–CMV–ssRFP–GFP–KDEL. For the construction of the plasmid pMRX–Halo–mGFP–Golgi, the Golgi sequence FLWRIFCFRK was inserted into the NotI/EcoRI site of pMRX–No–HaloTag7–mGFP–LC3–mRFP. For the selection of stable transformants by puromycin, the DNA fragments encoding puromycin N-acetyltransferase were inserted into the SalI site of the plasmid Halo–mGFP–Golgi. To generate the plasmids Halo–mGFP–YIPF3 and Halo–mGFP–YIPF4, the DNA fragments of human YIPF3 and YIPF4 were inserted into the NotI/EcoRI site of pMRX–No–HaloTag7–mGFP–Golgi, respectively.

## Cell culture

MEFs, HeLa Kyoto, HEK293T, and plat-E cells were cultured in DMEM (043-30085; Wako Pure Chemical Industries) supplemented with 10% fetal bovine serum, 1% L-glutamine, and 50 µg/mL penicillin–streptomycin in a 37 °C, 5% $CO_2$ incubator. The Plat-E cells were provided by T. Kitamura (The University of Tokyo,

Japan). For nutrient starvation, the cells were cultured in EBSS (E2888; Sigma-Aldrich). For the bafilomycin A$_1$ treatment, the cells were cultured with 125 nM bafilomycin A$_1$ (11038; Cayman Chemical). Transient transfections were carried out using Lipofectamine 2000 (11668-019; Invitrogen) and cells were used in experiments 24 h later.

For doxycycline treatment, HeLa cells stably expressing the plasmid Tet-On mRFP–EGFP Golgi were cultured with 2 μg/mL doxycycline (D3447; Sigma-Aldrich) for 24 h. For TMR-conjugated ligand (G825A; Promega) treatment, HeLa cells stably expressing Halo-based construct were cultured with 100 nM ligands for 20 min. After these treatments, cells were washed with PBS twice and incubated in DMEM supplemented with 10% fetal bovine serum, 1% L-glutamine, and 50 μg/mL penicillin–streptomycin or EBSS, followed by confocal microscopy or biochemical analysis.

## Virus production and generation of stable cell lines

Recombinant retroviruses were prepared as previously described (Saitoh et al, 2003). To generate stable cell lines, cells were cultured with recombinant retroviruses or lentiviruses and 10 μg/mL polybrene (H9268; Sigma-Aldrich), and they were selected in growth medium with puromycin (14861-71; InvivoGen) or blasticidin (029-18701; Wako Pure Chemical Industries).

## Establishment of KO cell lines by CRISPR–Cas9

YIPF3-KO, YIPF4-KO, and FIP200-KO HeLa cell lines were established using the below CRISPR guide RNAs (gRNAs). Annealed gRNA oligonucleotides were inserted into vector px458, and the gRNA construct was transfected into HeLa cells using ViaFect (Promega) transfection reagent. These cells were sorted into single cells by FACS into 96-well plates. The identity of candidate single-clone colonies was verified by immunoblotting using specific antibodies and genomic DNA sequencing. FIP200-KO HeLa cells were as described previously (Nakamura et al, 2020). The gRNA sequences were as follows: YIPF3, 5′-CCATTTCGGGCGCCGCCCGC-3′ and YIPF4, 5′-AAGGTGAAGTCCCCGTTAGT-3′.

## siRNA knockdown

The siRNAs for FIP200 and siControl were purchased from Sigma-Aldrich, and those for YIPF3, YIPF4, and CALCOCO1 were purchased from Thermo Fisher Scientific. The sequences are presented below. The siRNAs were transfected into HeLa cells stably expressing Halo–mGFP–Golgi using Lipofectamine RNAiMAX (13778-150; Invitrogen), in accordance with the manufacturer's instructions. The transfected cells were used for the subsequent experiment 72 h later. siYIPF3, 5′-AAUAGCAGCUGUGUGAGUCUUG-3′; siYIPF4, 5′-GCAGCCAAACUUUGAUUAAAGUU-3′; siCALCOCO1, 5′-GCACCAUAGCCGAACUACAdTdT-3′; siFIP200, 5′-GAUCUUAUGUAGUCGUCCAdTdT-3′; and siControl, 5′-UCGAAGUAUUCCGCGUACGdTdT-3′.

## Cell lysis and immunoprecipitation

Cells were rinsed twice with ice-cold PBS and lysed in a lysis buffer (50 mM Tris-HCl, pH 7.4, 150 mM NaCl, 1 mM EDTA, and 1%

Triton X-100) with phosphatase inhibitor cocktail (Roche), protease inhibitor cocktail (Roche) and 1 mM phenylmethylsulfonyl fluoride. For immunoprecipitation, the soluble fractions from the cell lysates were obtained after centrifugation at $20{,}400 \times g$ for 15 min. The protein concentration was measured using the Bradford assay. Each lysate was subjected to immunoprecipitation for 2 h or overnight with anti-FLAG M2 affinity agarose affinity gels (Sigma-Aldrich) or anti-HA magnetic beads (Thermo Fisher Scientific) with constant rotation at 4 °C.

After the incubation, the gels or beads were washed three times with ice-cold IP buffer (50 mM Tris-HCl, pH 7.4, 150 mM NaCl, 1 mM EDTA, and 1% Triton X-100) and washed with wash buffer (50 mM Tris-HCl, pH 7.4, and 150 mM NaCl). The samples were subsequently boiled in sample buffer (62.5 mM Tris-HCl, pH 6.8, 5% glycerol, 2% sodium dodecyl sulfate, 1% dithiothreitol, and bromophenol blue).

## In-gel fluorescence imaging and immunoblotting

After obtaining cell lysates, the soluble fractions were obtained after centrifugation at $600 \times g$ for 10 min. Samples were separated by sodium dodecyl sulfate-polyacrylamide gel electrophoresis (SDS-PAGE). For in-gel fluorescence imaging, the gel was immediately visualized using the ChemiDoc Touch MP imaging system (Bio-Rad) after SDS-PAGE. For immunoblotting, samples were transferred to polyvinylidene fluoride membranes. The membranes were blocked with 1% skim milk or 1% BSA in TBST, and incubated with primary antibodies diluted in blocking solution. The membranes were washed with TBST, incubated with HRP-conjugated secondary antibodies in blocking solution, and washed with TBST. The immunoreactive bands were detected using Immobilon Forte Western HRP substrate (WBLUF0500; Merck Millipore) or ImmunoStar LD (290-69904; Wako Pure Chemical Industries) on a ChemiDoc Touch MP imaging system (Bio-Rad). For the quantification of Golgiphagy activity, the band intensities of Halo and Halo-mGFP–Golgi in gel were measured, and the ratio of the cleaved Halo fluorescent signals to total Halo fluorescent signals (Halo:Halo–mGFP–Golgi) was calculated. For the quantification of protein levels by western blotting, the band intensities of each protein were measured using Fiji software (ImageJ; National Institutes of Health) (Schindelin et al, 2012), and normalized by the loading control α-tubulin.

## Immunofluorescence and microscopy

Cells were fixed with 4% paraformaldehyde (PFA) for 20 min at room temperature. They were then washed twice with PBS, permeabilized with 50 μg/mL digitonin–PBS for 10 min, blocked with 0.2% gelatin–PBS for 30 min, and subsequently incubated with each primary antibody in 0.2% gelatin–PBS for 1 h at room temperature. After washing with PBS three times, the cells were incubated with fluorescence-conjugated secondary antibodies in 0.2% gelatin–PBS for 1 h at room temperature. The samples were mounted using ProLong Diamond Antifade Mountant (P36961; Invitrogen), and observed with an FV3000 confocal microscope (Olympus) operated using FV31S-SW (version 2.3.1.163). The images were adjusted using Fiji software.

## TMT-based proteomic analysis

The brains of neonatal mice ($Atg5^{+/+}$ in triplicate, $Atg5^{-/-}$ in quadruplicate, and $Atg5^{-/-};NSE\text{-}Atg5$ in triplicate) were lysed in

500 μL of 6 M guanidine-HCl containing 100 mM Tris-HCl, pH 8.0, and 2 mM DTT. The lysates were dissolved by heating and sonication, followed by centrifugation at $20,000 \times g$ for 15 min at 4 °C. Proteins (100 μg each) were reduced in 5 mM DTT at room temperature for 30 min, alkylated in 27.5 mM iodoacetamide at room temperature for 30 min in the dark, and subjected to methanol/chloroform precipitation. After solubilization with 25 μL of 0.1% RapiGest SF (Waters) in 50 mM triethylammonium bicarbonate, the proteins were digested with 1 μg of trypsin/Lys-C mix (Promega) for 16 h at 37 °C. The peptide concentrations were determined using the Pierce quantitative colorimetric peptide assay (Thermo Fisher Scientific). Approximately 25 μg of peptides for each sample was labeled with 0.2 mg of TMT-10plex reagents (Thermo Fisher Scientific) for 1 h at 25 °C. After the reaction was quenched with hydroxylamine, all the TMT-labeled samples were pooled, acidified with trifluoroacetic acid (TFA), and fractionated using the Pierce high pH reversed-phase peptide fractionation kit (Thermo Fisher Scientific). Eight fractions were collected using 10%, 12.5%, 15%, 17.5%, 20%, 22.5%, 25%, and 50% acetonitrile (ACN). Each fraction was evaporated in a SpeedVac concentrator and dissolved in 0.1% TFA.

LC-MS/MS analysis of the resultant peptides (1 μg each) was performed on an EASY-nLC 1200 UHPLC connected to a Q Exactive Plus mass spectrometer through a nanoelectrospray ion source (Thermo Fisher Scientific). The peptides were separated on a 75 μm inner diameter × 150 mm C18 reversed-phase column (Nikkyo Technos) with a linear gradient of 4–20% ACN for 0–180 min and 20–32% ACN for 180–220 min, followed by an increase to 80% ACN for 220–230 min. The mass spectrometer was operated in a data-dependent acquisition mode with a top 15 MS/MS method. MS1 spectra were measured with a resolution of 70,000, an automatic gain control (AGC) target of 3e6, and a mass range of 375–1400 $m/z$. HCD MS/MS spectra were acquired at a resolution of 35,000, an AGC target of 1e5, an isolation window of 0.4 $m/z$, a maximum injection time of 100 ms, and a normalized collision energy of 32. Dynamic exclusion was set to 30 s. Raw data were directly analyzed against the SwissProt database restricted to *Mus musculus* using Proteome Discoverer version 2.2 (Thermo Fisher Scientific) with Mascot search engine version 2.5 (Matrix Science) for identification and TMT quantification. The search parameters were as follows: (a) trypsin as an enzyme with up to two missed cleavages; (b) precursor mass tolerance of 10 ppm; (c) fragment mass tolerance of 0.02 Da; (d) TMT of lysine and peptide N-terminus and carbamidomethylation of cysteine as fixed modifications; and (e) oxidation of methionine as a variable modification. Peptides were filtered at a false-discovery rate of 1% using the percolator node.

## Mice

All animal experiments were approved by the Institutional Animal Care and Use Committee of the University of Tokyo. *Atg5f/f;NSE-Atg5* mice were as previously described (Yoshii et al, 2016).

## Structure prediction, modeling, and presentation

AlphaFold-Multimer (preprint: Evans et al, 2022; Jumper et al, 2021) was used for structure prediction. Coot (Emsley et al, 2010) was used to introduce phosphoserines into the LIR of YIPF3.

Phenix (Liebschner et al, 2019) was used to apply molecular dynamics at 300 °C and subsequent geometry minimization to one of the resulting models. PyMOL (The PyMOL Molecular Graphics System, Version 2.0; Schrödinger, LLC) was used to compare and render the final model of the YIPF3 LIR bound to GABARAPL1 and the crystal structure of the TEX264 LIR bound to GABARAP (PDB accession code: 7VEC).

## Quantification and statistical analysis

All analyses were performed using GraphPad Prism 8.0. Statistical analyses were performed using one-way ANOVA followed by Tukey's multiple comparisons test. All quantitative data are denoted as mean ± standard deviation (SD). $P$ values ≤ 0.05 were considered significant ($*P \leq 0.05$; $**P \leq 0.01$; $***P \leq 0.001$, $****P \leq 0.0001$). No blinding was done.

## Data availability

The MS proteomics data have been deposited to the ProteomeXchange consortium via the jPOST partner repository with the dataset identifier PXD049220.

The source data of this paper are collected in the following database record: biostudies:S-SCDT-10_1038-S44318-024-00131-3.

## Peer review information

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

## Acknowledgements

The authors thank F Barr for providing anti-GRASP55 antibody, M Iwatani and N Beck for technical assistance, S Yamaoka for providing pMRX–IRES–puro and pMRX–IRES–bsr, N Mizushima and H. Chino for providing pCW57–CMV–ssRFP–GFP–KDEL, and T Johansen for providing pDest–mCherry–EYFP–ZDHHC17 and pDest–mCherry–EYFP–TMEM165. The authors also thank N Mizushima for supporting the initial stage of this study. TMT-based quantitative phosphoproteomics was performed under the Joint Usage and Joint Research Programs of the Institute of Advanced Medical

Sciences, Tokushima University. This work was supported by JST CREST (grant no. JPMJCR17H6 to TY), JSPS KAKENHI (grant no. 22H04982 to TY, 19K06637 and 23K05764 to AK, 25111005 to AK via Noboru Mizushima), and AMED (grant no. 22gm1410014h0001 to TY).

## Author contributions

**Shinri Kitta**: Conceptualization; Resources; Investigation; Visualization; Methodology; Writing—original draft. **Tatsuya Kaminishi**: Investigation; Visualization; Writing—original draft. **Momoko Higashi**: Investigation. **Takayuki Shima**: Investigation. **Kohei Nishino**: Data curation; Investigation. **Nobuhiro Nakamura**: Resources. **Hidetaka Kosako**: Data curation; Investigation. **Tamotsu Yoshimori**: Conceptualization; Supervision; Project administration; Writing—review and editing. **Akiko Kuma**: Conceptualization; Resources; Supervision; Investigation; Methodology; Writing—original draft; Project administration; Writing—review and editing.

Source data underlying figure panels in this paper may have individual authorship assigned. Where available, figure panel/source data authorship is listed in the following database record: biostudies:S-SCDT-10_1038-S44318-024-00131-3.

## Disclosure and competing interests statement

The authors declare no competing interests.

# Expanded View Figures

**A**

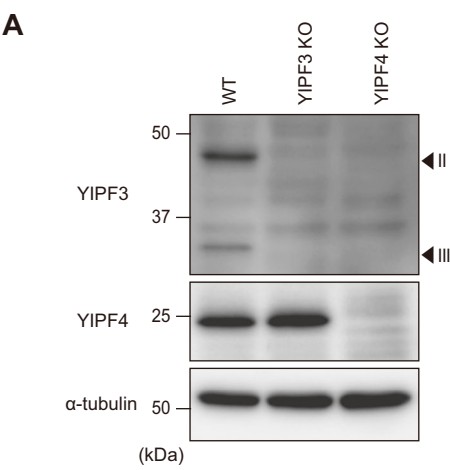

**B**

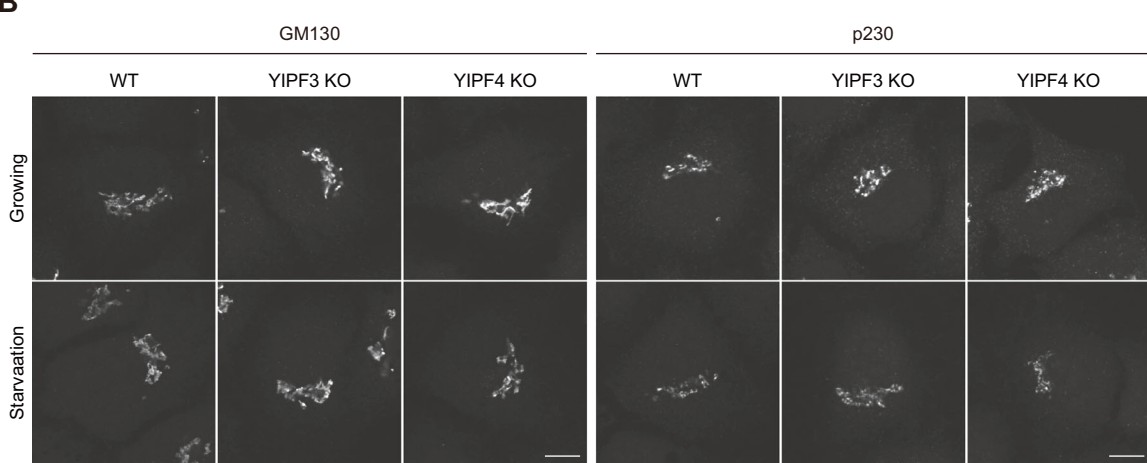

**Figure EV1.   Loss of YIPF4 suppresses YIPF3 protein levels.**

(**A**) Immunoblotting of indicated proteins in WT, YIPF3-KO, and YIPF4-KO HeLa cells. (**B**) Immunofluorescence image of WT, YIPF3-KO and YIPF4-KO HeLa cells. Cells were cultured in DMEM or EBSS (9 h) and stained with anti-GM130 or anti-p230 antibodies. Scale bars, 10 μm. Source data are available online for this figure.

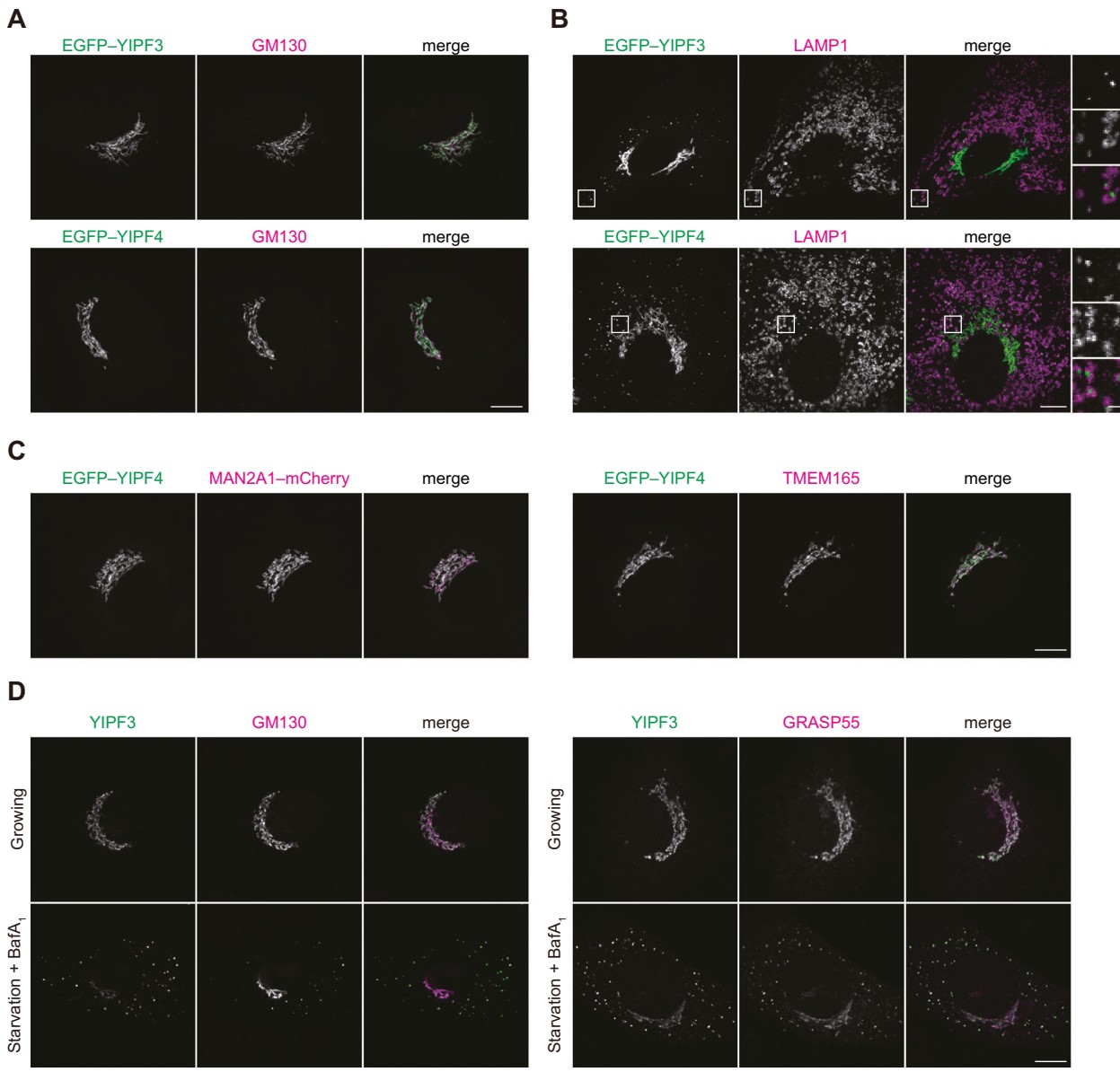

**Figure EV2.  EGFP–YIPF3 and EGFP–YIPF4 are localized to the Golgi apparatus under growing conditions, related to Fig. 2.**

(A) Immunofluorescence image of MEFs stably expressing EGFP–YIPF3 or EGFP–YIPF4. Cells were cultured in DMEM and stained with anti-GM130 antibody. Scale bars, 10 μm and 1 μm (insets). (B) Immunofluorescence image of MEFs stably expressing EGFP–YIPF3 or EGFP–YIPF4. Cells were cultured in EBSS (2 h) in the presence of 125 nM Baf A$_1$ and stained with anti-LAMP1 antibodies. Scale bars, 10 μm and 1 μm (insets). (C) Immunofluorescence image of MEFs stably expressing EGFP–YIPF4 or coexpressing EGFP–YIPF4 and MAN2A1–mCherry. Cells were cultured in DMEM. The indicated cells were stained with anti-TMEM165 antibody. Scale bars, 10 μm. (D) Immunofluorescence image of MEFs cultured in DMEM or EBSS (6 h) in the presence of 125 nM Baf A$_1$. Each cell was stained with anti-YIPF3 antibody and the indicated cells were stained with anti-GM130 (*cis*-Golgi) or anti-GRASP55 (*medial*-Golgi) antibodies. Scale bars, 10 μm. The ratio of YIPF3 puncta positive for each Golgi protein to total YIPF3 puncta was quantified in three independent experiments (mean ± SD). Ten cells were analyzed per condition in each experiment. Source data are available online for this figure.

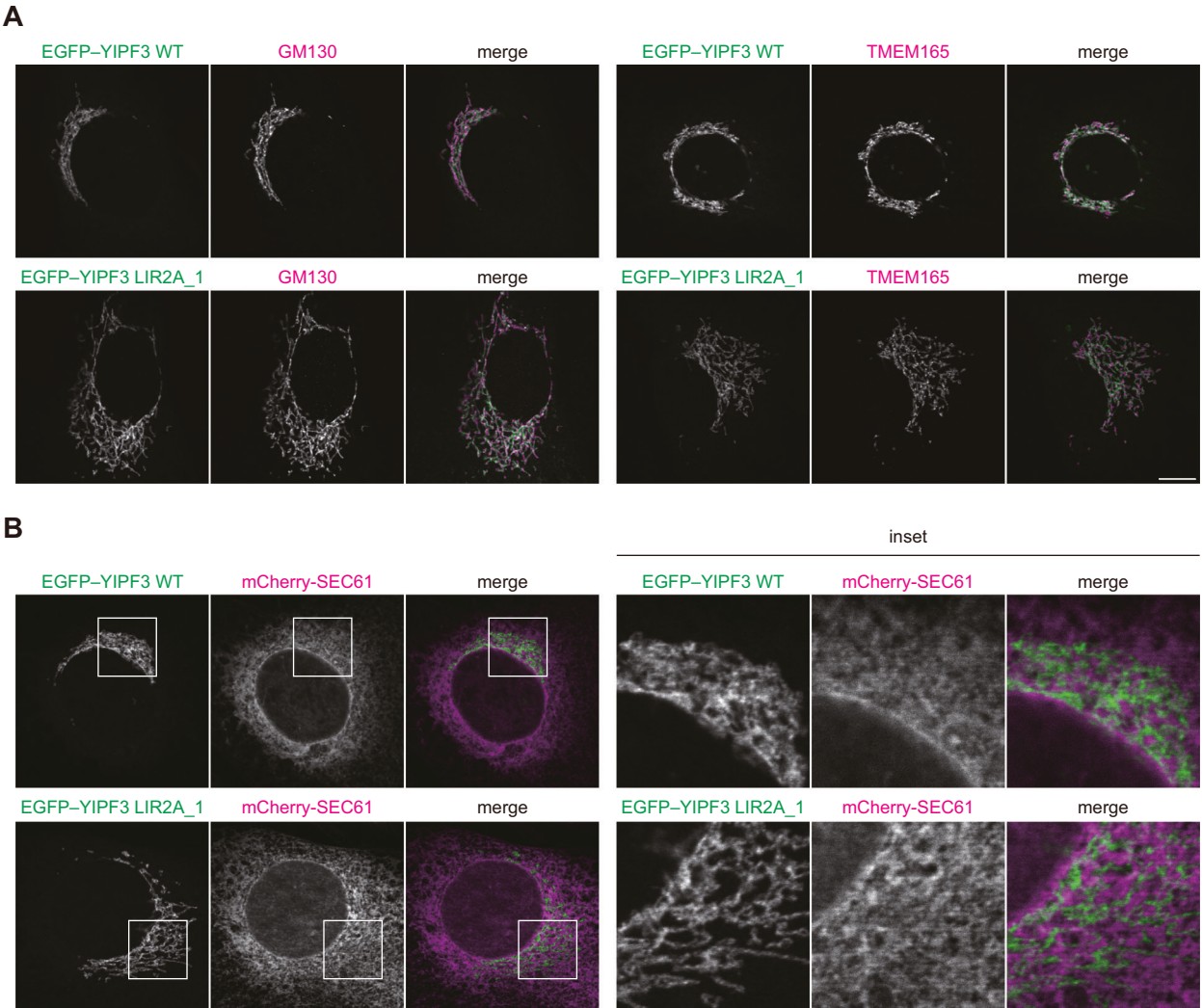

**Figure EV3. The Golgi apparatus is partially discontiguous and expanded in cells stably expressing YIPF3 LIR2A mutant.**

(A) Immunofluorescence image of MEFs stably expressing EGFP–YIPF3 WT or EGFP–YIPF3 LIR2A_1 under growing conditions. Cells were stained with anti-GM130 or anti-TMEM165 antibodies. Scale bars, 10 µm. (B) Immunofluorescence image of MEFs stably coexpressing EGFP–YIPF3 WT or EGFP–YIPF3 LIR2A_1 and mCherry-SEC61 under growing conditions. Scale bars, 10 µm and 2 µm (insets). Source data are available online for this figure.

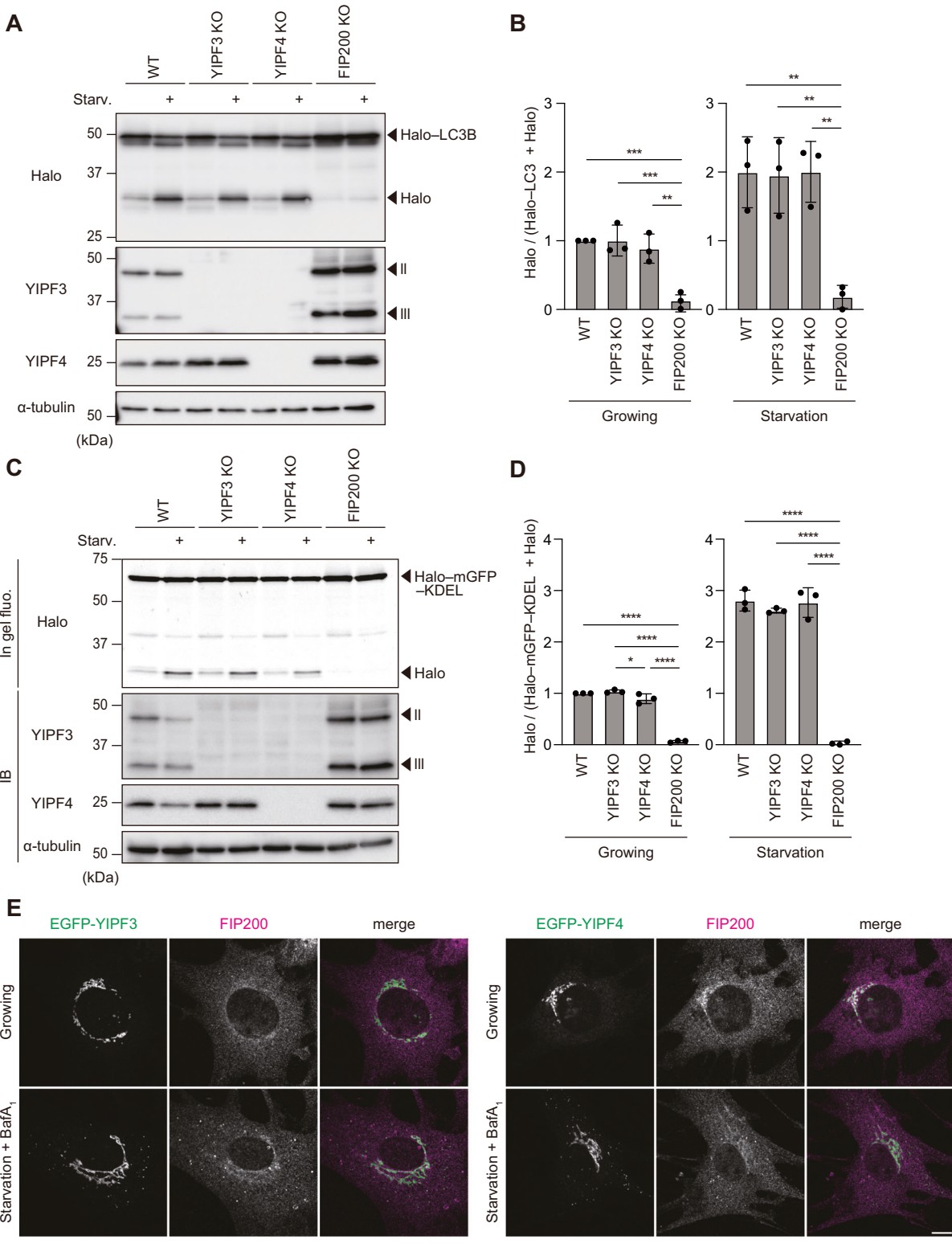

◀  **Figure EV4.  YIPF3 and YIPF4 are not involved in nonselective bulk autophagy and ER-phagy.**

(**A**) Immunoblotting of indicated proteins in WT, YIPF3-KO, YIPF4-KO, and FIP200-KO HeLa cells stably expressing Halo–LC3B. After cells were cultured in the presence of TMR-conjugated ligands for 20 min and washed out, they were cultured in DMEM or EBSS for 4 h. (**B**) Quantification of results shown in (**A**). The ratio of Halo:Halo–LC3B is shown. Each ratio is normalized to WT cultured in nutrient-rich medium. Error bars indicate mean ± SD ($n = 3$ independent experiments). (**C**) In-gel fluorescence image of Halo and immunoblotting of indicated proteins in WT, YIPF3-KO, YIPF4-KO, and FIP200-KO HeLa cells stably expressing Halo–mGFP–KDEL. After cells were cultured in the presence of TMR-conjugated ligands for 20 min and washed out, they were cultured in DMEM or EBSS for 9 h. (**D**) Quantification of results shown in (**C**). The ratio of Halo:Halo–mGFP–KDEL is shown. Each ratio is normalized to WT cultured in nutrient-rich medium. Error bars indicate mean ± SD ($n = 3$ independent experiments). (**E**) Immunofluorescence image of MEFs stably expressing EGFP–YIPF3 or EGFP–YIPF4 cultured in DMEM or EBSS in the presence of 125 nM Baf $A_1$ for 2 h. Cells were stained with anti-FIP200 antibodies. Scale bars, 10 μm. Data information: $P$ values were determined using one-way ANOVA with Tukey's multiple comparisons test (**B**, **D**). Symbols indicate: *$P \leq 0.05$; **$P \leq 0.01$, ***$P \leq 0.001$, ****$P \leq 0.0001$. Source data are available online for this figure.

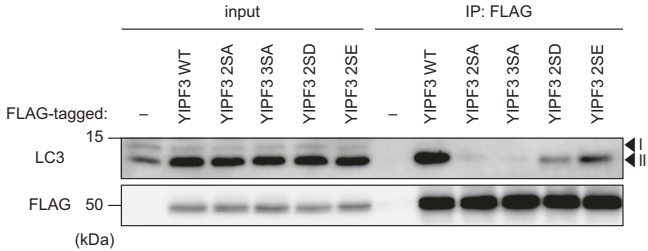

**Figure EV5.   YIPF3 phosphomimic mutants show reductions in their affinity to LC3.**

Immunoprecipitation of FLAG–YIPF3, FLAG–YIPF3$^{S45A, S46A}$ (2SA), FLAG–YIPF3$^{S43A, S45A, S46A}$ (3SA), FLAG–YIPF3$^{S45D, S46D}$ (2 SD) or FLAG–YIPF3$^{S45E, S46E}$ (2SE) in HEK293T cells. Cells were transfected with FLAG–YIPF3 WT or each serine mutants. The lysates were immunoprecipitated with anti-FLAG antibody and detected with anti-LC3 and anti-FLAG antibodies. Source data are available online for this figure.

