## [Peer Review File · The EMBO Journal]

YIPF3 and YIPF4 regulate autophagic turnover of the Golgi apparatus

Shinri Kitta, Tatsuya Kaminishi, Momoko Higashi, Takayuki Shima, Kohei Nishino, Nobuhiro Nakamura, Hidetaka Kosako, Tamotsu Yoshimori, and Akiko Kuma

Corresponding author(s): Akiko Kuma (kuma@fbs.osaka-u.ac.jp) , Tamotsu Yoshimori (tamyoshi@fbs.osaka-u.ac.jp)

Review Timeline:

Submission Date:	8th Aug 23
Editorial Decision:	19th Sep 23
Revision Received:	15th Feb 24
Editorial Decision:	14th Mar 24
Revision Received:	21st Apr 24
Accepted:	8th May 24

Editor: William Teale

Transaction Report:

Dear Dr Kuma,

Thank you for submitting your manuscript for consideration by the EMBO Journal. It has now been seen by three referees whose comments are shown below. As you will see, Referees 2 and 3 are generally positive about your work. However, Referee 1 raises a collection of technical concerns. Would you be available for a Zoom call next week to discuss whether you judge these concerns addressable? In the meantime, I will approach Referees 2 and 3 again to specifically ask for their input on Referee 1's points. Depending on the direction of our discussion, I will be able to make a decision on whether to invite revisions after we e-meet.

Yours sincerely,

William Teale

William Teale, PhD
Editor
The EMBO Journal
w.teale@embojournal.org

Referee #1:

Kitta et al present a characterization of the YIPF3/4 complex they found to be enriched in autophagy-impaired ATG5 mutant mouse brains. They went on to develop a model that these (cis-)Golgi proteins mediate selective Golgi turnover by autophagy via the specific binding of YIPF3 to a subset of phagophore/autophagosome-associated ATG8 family proteins. They also proposed that putative phosphosites on YIPF3 are important for this interaction in case of overexpressed proteins. Unfortunately, the study suffers from a range of problems including very weak effects seen in loss-of-function tests on Golgi turnover (not necessarily visible when looking at western blots and not even always statistically significant depending on which control genotypes are used for comparisons), reliance on overexpressed proteins in most tests, preliminary analysis of YIPF3 phosphorylation, overinterpretations, and complete lack of a hint whether this proposed Golgi turnover mechanism has any in vivo relevance. These issues dramatically reduce my enthusiasm towards this work.

Major comments

1. In general, the effects of YIPF3/4 loss on Golgi turnover are minor, especially when compared to an already characterized Golgiphagy receptor (for example, see the level of upregulation of the Golgi protein GM130 in GMAP LIR mutants in Fig. 5 of Rahman et al, 2022 Cell Rep). In the most quantitative Halo-tag flux WB assays, the difference of Golgi turnover between knockdown and WT cells is rather small (Fig 6D, E) - the authors falsely claim in the text that Halo fluorescence is reduced "TO" 22-32% because it is not true: these are reduced "BY" 22-32%. I also see a stronger processed Halo band in fed YIPF3 KO cells compared to fed WT cells in Fig. 6B, which is at odds with the quantification in Fig 6C. Along these lines, the loading control tubulin band is just as much weaker for starved YIPF3 cells as Halotag signal is, compared to starved WT in the same panel. Is this a representative blot? If yes, then I'm sceptical about the accuracy of quantifications. In addition, the authors should not try to oversell their story by claiming 50% reduction of Golgiphagy flux upon YIPF3/4 loss in the discussion based on microscopy because the extent of change depends on the type of assay used (that is, it is much smaller in the more quantitative and reliable in-gel fluorescence of free Halotag).

2. Similarly, the differences in case of YIPF3/4 overexpression on quantitative Halo-based Golgi turnover are very small, it is not even statistically significant when compared to YIPF3 expression alone (which does not elevate YIPF3/YIPF4 levels). I don't think that enhanced Golgiphagy can be claimed if it is only seen when compared to certain but not all control genotypes.

3. A related note is that the authors claim in-gel fluorescence of Halotag to be the most quantitative assay, but its advantage is hindered by normalizing the signal to the loading control tubulin, which is seen as highly saturated bands on conventional western blots. It would be important to improve the reliability of quantifications for all Halo-flux assays by repeating anti-tubulin westerns using a dilution series (say, 2x 4x 10x etc) because this would give a much better estimate of tubulin levels in each sample.

4. While reduced, there is still an obvious turnover of YIPF proteins in FIP200-KO cells (unlike in Baf treated cells) in Fig 3D-G. This Halo-flux assay should be repeated in other autophagy knockout cells (such as Atg5-KO, Atg14L-KO) to reveal how this residual protein breakdown happens. Is it a partially FIP200-independent or canonical autophagy-independent process?

5. It is disappointing that too many tests and claims rely on overexpressed proteins only. Most importantly, do endogenous YIPF3 and YIPF4 localize to cis, medial, and trans-Golgi? It would explain the weak phenotypes if these were only involved in the turnover of a subset of Golgi stacks. Do endogenous YIPF3 and YIPF4 accumulate if autophagy is blocked in cells? By the

way, where is the statistics for brain proteomic data? Do YIPF3/4 proteins colocalize with an endogenous medial Golgi marker? Is there an interaction between endogenous YIPF3 and ATG8 family proteins? Etc.

6. Regulation of YIPF3 interaction with ATG8 family proteins by phosphorylation is potentially interesting but very preliminary. First, direct evidence for the phosphorylation of S45 and/or S46 is lacking. Second, this aspect of the work is incomplete without identifying the kinase(s) involved, which might give a hint to the in vivo importance of this molecular interaction.

7. "Statistical analyses were performed using one-way ANOVA followed by Tukey's multiple comparisons test." ANOVA is only suitable for the comparison of datasets that all show normal, Gaussian distribution. This is often not the case when values are close to zero (e.g., Figs. 2B, 5D, EV4B, D). Please include a supplementary table for the statistics containing the raw data, averages, standard deviations, results of data normality tests, the type of statistical test selected based on this (ANOVA versus a non-parametric test), and actual p values.

Minor comments

8. Statistical analysis for WB data in Fig 3A is missing. The "trends" in protein level changes cannot be trusted without that.

9. Endogenous YIPF3 binding to FLAG-LC3B is barely visible, and FLAG-YIPF3 interaction with endogenous GABARAPL1 is missing from Fig 4. Please correct.

10. There are many inconsistencies in the text. For example:

"how the Golgi is recognized by the autophagosome" and "In mammals, the organelles to be degraded by recognized directly or indirectly by ATG8 family proteins (LC3/GABARAP family proteins) on autophagosomes" - please change these to "forming autophagosome" (or phagophore), because the autophagosome is a closed double-membrane vesicle that no longer recognizes/captures cargo, this happens before the completion of autophagosome formation. Organelles to be degraded are recognized by ATG8 family proteins not only in mammals, as there are many examples from invertebrates, yeast etc. This is an evolutionarily conserved mechanism so please delete "In mammals".

"Golgi apparatus was also reported to be a substrate for autophagy" - references are missing from the introduction.

"We did not observe the fragmentation of Golgi in HeLa cells during starvation" - please change to "in MEF or HeLa cells"

"Upon starvation, the amounts of YIPF3 (forms II and III) and YIPF4 were found to be reduced in a time-dependent manner, and these reductions were abolished in FIP200-KO cells." - please refer to Fig 3A here

"the YIPF3 LIR mutant showed an elongated Golgi morphology" - this sentence is not to the point, please rephrase to „overexpression of YIPF3 LIR mutant led to an expansion of Golgi"

Referee #2:

In their manuscript 'Golgi membrane proteins YIPF3 and YIPF4 regulate turnover of the Golgi apparatus through autophagy', Kuma and colleagues identified the YIPF3/YIPF4 complex to be a novel receptor for selective autophagy of the Golgi. The authors convincingly showed that both proteins are required for the efficient turnover of the Golgi. In addition, the authors identified an essential LIR motif within YIPF3 that is regulated via two adjacent phospho-sites. The findings are novel and of great interest for the autophagy field. The developed tools are valuable additions and will be very useful for future studies on this topic. The experimental setup is of high quality and as such I find the provided manuscript suitable for publication in EMBO J.

I have no major concerns regarding the main message of the manuscript.

Minor concerns that should be addressed:

Please use correct nomenclature for mouse proteins vs human protein (e.g. Fig 1A).

Proteomics: Please provide a searchable excel file listing all significantly up or down regulated proteins, columns with respective p-values/fold change, and a column highlighting golgi-resident proteins.

'We did not observe the fragmentation of Golgi in HeLa cells during starvation; however, with bafilomycin A1 (Baf A1), a vacuolar ATPase inhibitor, we observed the significant appearance of punctate structures of EGFP-YIPF3 and EGFP-YIPF4 in the cytoplasm, in addition to the ribbon-like Golgi structure (Fig 2A and B).' 1. The logic (start of the paragraph) as well as figure legends indicate that experiments were performed in MEFs, not HeLa. Please comment / correct. 2. While images are convincing regarding the general message (more than in growing conditions), number of puncta in Fig. 2A/C (EBSS+Baf) seem to be higher than in the bar quantification (2B). Was counting done manually or semi-automatically (automated background, automated numbers /cell)? Can this be presented more intuitively (e.g. adding an image with a mask in EV Fig so that reader understands which dots were counted and which not and why)?

,The processing of Halo-mGFP-YIPF3 or Halo-mGFP-YIPF4 was suppressed by the deletion of FIP200, showing that YIPF3 and YIPF4 were degraded by autophagy upon starvation (Fig 3 D-G). ' change to 'Increased processing ... upon starvation...'
Clearly, constructs were processed also in FIP200 KO to some extent.

Fig 4B & text: based on the provided data, endogenous YIPF3 did NOT bind to FLAG-tagged LC3B, but only to FLAG-tagged GABARAP and GABARAPL1. Please correct.

Endogenous LC3B did seem to interact with FLAG-tagged YIPF3. Maybe a point for discussion. Interesting also in respect of recent finding that N-term of LC3B may interact with membrane (Tooze lab, PMID: 37288820).

Fig EV3: Reminds a bit on ER - please add an ER marker in parallel (for LIR mutant) to see the impact / correlation.

Fig 5C & text: since the authors used purple to visualize the mRFP part, there are no red-only puncta. Please change or adapt the text to avoid confusions. Based on the provided images, KO of YIPF3,4 does change the morphology upon starvation (similar to FIP200 KO). Please clarify.

Fig 6B, 4th lane, Halo blot: The quantification shows a drop in YIPF3 KO cells. However, in contrast to the WT in lane 1, there is a clearly visible Halo signal in YIPF3 KO cells. As I understood the assay, this would suggest a higher activity in this specific experiment. Please explain.

Fig 6B: FIP200 levels strongly dropped in starved YIPF4 KO cells. Was this consistent in all experiments?

Fig.6E: change label siYIPF4 for siYIPF3/4

'... (Fig 6B). This confirmed that the appearance of free Halo tag was dependent on autophagy and lysosomal degradation. These findings showed that the mRFP-EGFP-Golgi reporter can be used to measure Golgiphagy flux quantitatively by SDS-PAGE.' I assume the authors meant Halo-mGFP-Golgi reporter.

Text referring to Fig 6B,C: the note (equivalent to YIPF3/4 DKO) is inconsistently used in comparison to previous paragraphs and therefore implies that this specific experiment was also done in a DKO line.

'Indeed, CALCOCO1 was recently reported to be a soluble Golgiphagy adaptor.' Please change to receptor/co-receptor for consistency and add the respective reference.

7A, B: The quantification of specifically YIPF3 levels presented, do not seem to correlate with the presented Immuno blot (YIPF3 antibody). I understood that the total of all YIPF3 signals (endogenous + exogenous) were taken into account. However, already the signal of HA-YIPF3 in lane 3,4 (by eye) surely more than compensates the decrease of the endogenous protein. And in lane 7,8 a 3-fold increase seems far too little. Maybe for the stabilization of the endogenous protein. The signal of the exogenous protein seems to be 10x more. Please reconfirm the given quantification and potentially correct fig & text.

As this is the only overexpression condition where Golgiphagy is increased, please provide sample images of the Golgi in Ext Fig. for YIPF3 / YIPF4 co-expressing cells in extended figures.

Fig8: By eye, the Halo signal in lane 6 seems to be stronger than in lane 2, which would indicate a small overcompensation. This is not visible in the quantification, which show a non-significant reduction of activity compared to WT cells. While of no relevance for the message of the paper, it is irritating that blot and quantification does not seem to correlate.

Non-essential suggestions:

Fig 3B - squares and circles were difficult to identify in the printed pdf. Please use more different forms, e.g. empty triangle and filled circle.

Fig 8A: I would suggest to remove the labels on the left side of the YIPF3 blot. Form I and III of HA-YIPF3 runs on too similar height as form II and I of the endogenous protein, respectively. The info is not essential and on the first sight one gets the impression that there is suddenly remaining endogenous protein in the KO cells in lane 5-8.

Given the previous report, the finding of the authors that CALCOCO1 KD did not decrease Golgiphagy in HeLa cells under basal as well as starvation is unexpected. Please comment on this contradictory finding.

Referee #3:

Kitta et al report that the two five-transmembrane proteins YIPF3 and -4 located in the cis-Golgi act together as a Golgiphagy receptor. YIPF3 binds to ATG8 family proteins via a LIR motif which is regulated by phosphorylation. YIPF3 and -4 form a complex. YIPF4 does not bind to ATG8s but is required for the stability of YIPF3 so that these two transmembrane proteins together act as a Golgiphagy receptor upon starvation dependent on the LIR motif in YIPF3. They become degraded in a Bafilomycin A1-sensitive manner upon starvation and Halo-tagged YIPF3 or -YIPF4 constructs accumulate in the lysosome upon starvation. The authors also developed a general Golgiphagy reporter as a tandem tag and as a Halo-tag construct containing a 10-amino acid sequence shown earlier to act as a Golgi targeting motif. Using this reporter and KO cells they show that the YIPF3/-4 complex is required for efficient Golgiphagy upon starvation. Overexpression of the complex also increased Golgiphagy while reconstitution of KO cells with a YIPF3 LIR mutant could not rescue starvation induced Golgiphagy. Expression of the YIPF3 LIR mutant showed an elongated Golgi morphology suggesting that YIPF3/-4-mediated Golgiphagy is involved in maintaining normal Golgi morphology. The authors show that YIPF3 is phosphorylated on serine using a pSer Ab following IP of FLAG-tagged YIPF3 from cells. Immediately N-terminal to the core LIR motif (FEEDM) are two Ser residues and mutation of these to Ala residues inhibited binding of LC3 to YIPF3 and did also inhibit starvation induced Golgiphagy. This is a very interesting study of high quality reporting the first membrane-spanning Golgiphagy receptor complex and also specific reporters for monitoring Golgiphagy. The study is very well performed with good controls yielding convincing results, and where relevant quantifications of results have been performed.

I have only one major critical point to make that would strengthen the story if addressed.

1. The authors claim that phosphorylation of the LIR motif in YIPF3 is required for YIPF3-ATG8 interaction and for YIPF3-YIPF4-mediated Golgiphagy.

- a) The authors show that YIPF3 can be phosphorylated on serine, but it is not shown that the S45 and/or S46 residues are phosphorylated. A control where the S45A/S46A mutant YIPF3 is immunoprecipitated after expression in YIPF3 KO cells and blotted with pSer Ab is lacking from Fig 9B. Did the authors try this experiment? It is highly likely that other sites in YIPF3 is phosphorylated so that the result would still be that a band is seen with the pSer Ab, but if the band disappears or is weakened significantly it would be suggestive of these sites being targets of phosphorylation. Ideally, it would be important to see if phospho-proteomics experiments would show if S45/S46 are phosphorylated and if this is induced upon starvation.
- b) The authors could test Asp or Glu mutants of S45 and S46 and test if these induce Golgiphagy when expressed in YIPF3 KO cells and also do binding experiments to show if the S45D/S46D mutant (or E mutants) binds more strongly to the ATG8s in pull down assays or IPs than the WT YIPF3.
- c) Do the authors have any clue which protein kinases(s) are involved in phosphorylating S45 and/or S46. This could be tested using chemical inhibitors of some relevant kinases.

Exploring the phosphorylation part more will strengthen this paper. If this is not done, the authors need to moderate their conclusions regarding the role of phosphorylation here and discuss this in the Discussion section.

Minor point:

No binding to LC3s is seen in the IP in Figure 4B while a strong binding is seen to GABARAPL1 and a weaker binding to GABARAP. However, in Figure 4C LC3 is IPed very efficiently it seems. How do the authors explain this?

Also, the input of the different FLAG-tagged ATG8as vary a lot in Figure 4B.

It is also unexpected that GABARAP binds seemingly much more poorly than GABARAPL1 as these are very similar and usually show similar binding to a multitude of LIR-containing proteins. Why this difference here. It would be nice to see this experiment repeated with a more equal input and some quantification of the binding.

Response to the reviewer comments:

We would like to thank for the reviews of our manuscript # EMBOJ-2023-115254 entitled "Golgi membrane proteins YIPF3 and YIPF4 regulate turnover of the Golgi apparatus through autophagy ". We appreciate the reviewer's constructive and valuable comments which improve our manuscripts. Below we have answered all the comments made by the reviewers. The revisions are noted in colored letters in the manuscript. We hope that our responses are reasonable and acceptable for the reviewers.

Referee #1:

Kitta et al present a characterization of the YIPF3/4 complex they found to be enriched in autophagy-impaired ATG5 mutant mouse brains. They went on to develop a model that these (cis-)Golgi proteins mediate selective Golgi turnover by autophagy via the specific binding of YIPF3 to a subset of phagophore/autophagosome-associated ATG8 family proteins. They also proposed that putative phosphosites on YIPF3 are important for this interaction in case of overexpressed proteins. Unfortunately, the study suffers from a range of problems including very weak effects seen in loss-of-function tests on Golgi turnover (not necessarily visible when looking at western blots and not even always statistically significant depending on which control genotypes are used for comparisons), reliance on overexpressed proteins in most tests, preliminary analysis of YIPF3 phosphorylation, overinterpretations, and complete lack of a hint whether this proposed Golgi turnover mechanism has any *in vivo* relevance. These issues dramatically reduce my enthusiasm towards this work.

Major comments

1. In general, the effects of YIPF3/4 loss on Golgi turnover are minor, especially when compared to an already characterized Golgiphagy receptor (for example, see the level of upregulation of the Golgi protein GM130 in GMAP LIR mutants in Fig. 5 of Rahman et al, 2022 Cell Rep).

Response: We thank the reviewer for raising the important point. As the reviewer pointed, GM130 strongly accumulates in GAMP LIR mutant flies in the previous report (Cell Rep. 2022). On the other hand, in HeLa cells, GM130 accumulates only 1.5-fold even in FIP200-KO cells compared to WT cell as shown in Fig.3A. We assume that this is due to the difference between *in vivo* experiments and cultured cell experiments. In general, autophagic substrates accumulate more *in vivo* than in cultured cells when autophagy is suppressed, because the accumulated substrates are diluted by cell division in cultured cells. This may explain why the stronger accumulation of GM130 was observed in flies than cultured cells. Alternatively, this could be due to the differences in assays (LIR mutant or

knockout) and species.

We observed a 35 to 60 % reduction in Golgiphagic activity in YIPF4 knockout cells in both imaging and biochemical analysis using our probes (RFP-EGFP-Golgi in Fig.5 and Halo-mGFP-Golgi in Fig.6). Therefore, we think that the effects of YIPF3/4 loss on Golgi turnover are partial, but not minor. This partial effect may be due to the presence of other receptors for Golgiphagy besides YIPF3/4, as in the case of ER-phagy.

In the most quantitative Halo-tag flux WB assays, the difference of Golgi turnover between knockdown and WT cells is rather small (Fig 6D, E) -the authors falsely claim in the text that Halo fluorescence is reduced "TO" 22-32% because it is not true: these are reduced "BY" 22-32%.

Response: We apologize for the mistake. We have corrected "to" to "by" in the text.

In Fig.6D and E, YIPF3 and YIPF4 were deleted by knockdown. Although the knockdown efficiency is good, there are still a few YIPF3 and YIPF4 bands remaining. This probably explains why the effect of YIPF3/4 knockdown looks rather small (a 20-32% reduction). We also evaluated the Golgiphagy activity in YIPF knockout cells in Fig.6B and C. Indeed, Golgiphagic activity was more suppressed in YIPF4-knockout cells (a 36-48% reduction) than knockdown cells in the Halo-mGFP-Golgi assay. Based on these results, we think the involvement of YIPF3/4 in Golgiphagy is not small. It may be even rather large, albeit partial.

I also see a stronger processed Halo band in fed YIPF3 KO cells compared to fed WT cells in Fig. 6B, which is at odds with the quantification in Fig 6C.

Response: Thank you for pointing this out. This is because the expression level of the probe is higher in YIPF3-KO cells than WT cells (1.7-fold) in the previous manuscript, resulting in more cleaved Halo tag bands in appearance. For the quantification of Golgiphagy activity, the cleaved Halo-tag band was divided by the total Halo-tag band [Halo/(Halo-mGFP-Golgi+Halo)]. This gave the unfavorable impression that there was no correlation between the appearance and the quantitative results. To remedy this problem, we have sorted cells expressing the same level of the probes and redid the experiments using these cells in the revised manuscript.

Along these lines, the loading control tubulin band is just as much weaker for starved YIPF3 cells as Halotag signal is, compared to starved WT in the same panel. Is this a representative blot? If yes, then I'm sceptical about the accuracy of quantifications.

Response: For the quantification, we don't use tubulin for the loading control. We evaluate the Golgiphagic activity by the ratio of the cleaved Halo fluorescent signals to total Halo fluorescent signals detected in gels. This quantification method has been reported by

Mizushima's group (eLife 2022, PMID: 35938926). We apologize for the confusion, and will clarify this in the method section.

In addition, the authors should not try to oversell their story by claiming 50% reduction of Golgiphagy flux upon YIPF3/4 loss in the discussion based on microscopy because the extent of change depends on the type of assay used (that is, it is much smaller in the more quantitative and reliable in-gel fluorescence of free Halotag).

Response: We apologize that we put the incorrect figure number in the sentence in the first paragraph of Discussion, line 7,8 from the bottom, "*We observed that the deletion of YIPF3 or YIPF4 resulted in a nearly 50% decrease in Golgiphagy flux, as judged by our Halo-mGFP-Golgi reporter assay (Fig 5D)*". The correct figure is Fig.6C. We evaluated the effects of YIPF3/4 loss on Golgiphagy in YIPF4-KO cells by microscopic analysis (Fig.5D) and the Halo-tag analysis (Fig.6C). In the both assays, we observed more than 35% reduction in the activity of Golgiphagy in YIPF4-KO cells compared to WT cells, which is partial, but not small. We have described that "*We observed that the deletion of YIPF3 or YIPF4 resulted in more than 35% decrease in Golgiphagy flux,*" in the revised manuscript.

Microscopic analysis in Fig.5D

53% reduction in growing and 68% reduction in starvation conditions in YIPF4 KO cells compared to WT respectively.

Halo-Golgi assay in Fig.6C

48% reduction in growing and 36% reduction in starvation conditions YIPF4 KO cells compared to WT respectively

2. Similarly, the differences in case of YIPF3/4 overexpression on quantitative Halo-based Golgi turnover are very small, it is not even statistically significant when compared to YIPF3 expression alone (which does not elevate YIPF3/YIPF4 levels). I don't think that enhanced Golgiphagy can be claimed if it is only seen when compared to certain but not all control genotypes.

Response: We agree that the effect of YIPF3/4 overexpression on Golgi turnover is small, but it is statistically significant when compared to overexpression of WT or YIPF4 alone in Fig.7C. As the reviewer pointed out, it is not statistically significant only when compared to YIPF3 expression alone. Therefore, we have repeated the experiment (n=6). Although no significant difference was obtained, there was a tendency that there may be a difference

(the $p=0.0552$ under growing condition, $p=0.0635$ under starvation conditions). One possible explanation would be that overexpression of YIPF3 alone might slightly affect Golgiphagic activity because it has LIR motif, and this may make it difficult to find significant differences.

3. A related note is that the authors claim in-gel fluorescence of Halotag to be the most quantitative assay, but its advantage is hindered by normalizing the signal to the loading control tubulin, which is seen as highly saturated bands on conventional western blots. It would be important to improve the reliability of quantifications for all Halo-flux assays by repeating anti-tubulin westerns using a dilution series (say, 2x 4x 10x etc) because this would give a much better estimate of tubulin levels in each sample.

Response: For the quantification, we don't use tubulin for the loading control. We evaluate the Golgiphagy activity by the ratio of the processed Halo-fluorescent signals to total Halo-fluorescent signals detected in gels. This quantification method has been reported by Mizushima's group (eLife 2022, PMID: 35938926). We apologize for the confusion. We clarified this in the method section.

4. While reduced, there is still an obvious turnover of YIPF proteins in FIP200-KO cells (unlike in Baf treated cells) in Fig 3D-G. This Halo-flux assay should be repeated in other autophagy knockout cells (such as Atg5-KO, Atg14L-KO) to reveal how this residual protein breakdown happens. Is it a partially FIP200-independent or canonical autophagy-independent process?

Response: We thank the reviewer for this suggestion. We have evaluated the Golgiphagic activity in ATG7-KO, ATG-14KO, ATG-16KO cells. The data suggested that residual protein breakdown seems to be autophagy-independent process.

Fig R1. In-gel fluorescence image of Halo and immunoblotting in HeLa WT and ATG-KO cells stably expressing Halo-mGFP-YIPF3. After cells were cultured in the presence of TMR-conjugated ligands for 20 min and washed out, they were cultured in DMEM or EBSS in the presence or absence of 125 nM BafA1 for 9 h.

For the quantification, the ratio of Halo:Halo-mGFP-YIPF3 was calculated. Each ratio was normalized to WT cultured in nutrient-rich medium. Error bars indicate mean \pm SD (n = 4). Statistical significance was determined using one-way analysis of variance (ANOVA) with Dunnett's posttest.

5. It is disappointing that too many tests and claims rely on overexpressed proteins only. Most importantly, do endogenous YIPF3 and YIPF4 localize to cis, medial, and trans-Golgi? It would explain the weak phenotypes if these were only involved in the turnover of a subset of Golgi stacks. Do endogenous YIPF3 and YIPF4 accumulate if autophagy is blocked in cells?

Response: We have tested several antibodies to show the endogenous behavior of YIPF3/4 and the Golgi proteins. Fortunately, anti-YIPF3, anti-GM130 (cis), and GRASP55 (medial) antibodies worked for IF and we have confirmed that endogenous YIPF3 were well co-localized with cis- and medial-Golgi. We could not perform IF with the combination of YIPF3 and TMEM165 (trans) because both are rabbit antibodies. We added the IF data in Fig.EV2D.

By the way, where is the statistics for brain proteomic data?

Response: We apologize. The MS proteomics data have been deposited to the ProteomeXchange consortium via the jPOST partner repository with the dataset identifier PXD049220. We have also provided a excel file of the data.

Do YIPF3/4 proteins colocalize with an endogenous medial Golgi marker? Is there an interaction between endogenous YIPF3 and ATG8 family proteins? Etc.

Response: We have performed IF experiments and confirmed that endogenous YIPF3 colocalized with endogenous medial Golgi marker (GRASP55). We added the data in Fig.EV2D. As for IP experiment, it is difficult to examine the endogenous-endogenous protein interaction. We showed the interaction between endogenous YIPF3 and exogenous ATG8s and vice versa, which we think is a general approach.

6. Regulation of YIPF3 interaction with ATG8 family proteins by phosphorylation is potentially interesting but very preliminary. First, direct evidence for the phosphorylation of S45 and/or S46 is lacking. Second, this aspect of the work is incomplete without identifying the kinase(s) involved, which might give a hint to the in vivo importance of this molecular interaction.

Response: We thank the reviewer for raising this point. This comment is the same as reviewer #3's. We have performed phospho-proteomics experiments to show if Ser45/Ser46 are phosphorylated, but we could not detect the phosphorylation. Instead, we have observed clear dropdown of phosphorylation in YIPF3 S45A/S46A mutant by IP using anti-pSer antibody, suggesting these sites are targets of phosphorylation. The results are very suggestive but still indirect, therefore, we have tone down our claim about the regulation of YIPF3 by phosphorylation in the revised manuscript (describe as putative phosphorylation sites etc.). We added the data in Fig.9B.

We agree that identification of the kinases is important, however, it will be a lot of work and we would like to address this issue in the future. We discuss this in the Discussion section (page 16, line 351).

7. "Statistical analyses were performed using one-way ANOVA followed by Tukey's multiple comparisons test." ANOVA is only suitable for the comparison of datasets that all show normal, Gaussian distribution. This is often not the case when values are close to zero (e.g., Figs. 2B, 5D, EV4B, D). Please include a supplementary table for the statistics containing the raw data, averages, standard deviations, results of data normality tests, the type of statistical test selected based on this (ANOVA versus a non-parametric test), and actual p values.

Response: The dots in the graph are the average of one experiment (counted more than 60 cells each experiment). Therefore, we think we can assume and calculate as Gaussian distribution following central limit theorem. We will include a supplementary table as Source Data.

Minor comments

8. Statistical analysis for WB data in Fig 3A is missing. The "trends" in protein level changes cannot be trusted without that.

Response: Following the reviewer's comments, we have avoided statistical statements in Fig 3A.

9. Endogenous YIPF3 binding to FLAG-LC3B is barely visible, and FLAG-YIPF3 interaction with endogenous GABARAPL1 is missing from Fig 4. Please correct.

Response: We apologize for missing the data. We added the data showing the interaction of

FLAG-YIPF3 with endogenous GABALAPL1 in Fig. 4.

10. There are many inconsistencies in the text. For example:

"how the Golgi is recognized by the autophagosome" and "In mammals, the organelles to be degraded by recognized directly or indirectly by ATG8 family proteins (LC3/GABARAP family proteins) on autophagosomes" - please change these to "forming autophagosome" (or phagophore), because the autophagosome is a closed double-membrane vesicle that no longer recognizes/captures cargo, this happens before the completion of autophagosome formation. Organelles to be degraded are recognized by ATG8 family proteins not only in mammals, as there are many examples from invertebrates, yeast etc. This is an evolutionarily conserved mechanism so please delete "In mammals".

"Golgi apparatus was also reported to be a substrate for autophagy" - references are missing from the introduction.

"We did not observe the fragmentation of Golgi in HeLa cells during starvation" - please change to "in MEF or HeLa cells"

"Upon starvation, the amounts of YIPF3 (forms II and III) and YIPF4 were found to be reduced in a time-dependent manner, and these reductions were abolished in FIP200-KO cells." - please refer to Fig 3A here

"the YIPF3 LIR mutant showed an elongated Golgi morphology" - this sentence is not to the point, please rephrase to „overexpression of YIPF3 LIR mutant led to an expansion of Golgi"

Response: We thank the reviewer for correcting the words and phrases. We corrected them as the reviewer suggested.

Referee #2:

In their manuscript 'Golgi membrane proteins YIPF3 and YIPF4 regulate turnover of the Golgi apparatus through autophagy', Kuma and colleagues identified the YIPF3/YIPF4 complex to be a novel receptor for selective autophagy of the Golgi. The authors convincingly showed that both proteins are required for the efficient turnover of the Golgi. In addition, the authors identified an essential LIR motif within YIPF3 that is regulated via two adjacent phospho-sites. The findings are novel and of great interest for the autophagy field. The developed tools are valuable additions and will be very useful for future studies on this topic. The experimental setup is of high quality and as such I find the provided manuscript suitable for publication in EMBO J.

I have no major concerns regarding the main message of the manuscript.

Minor concerns that should be addressed:

Please use correct nomenclature for mouse proteins vs human protein (e.g. Fig 1A).

Response: Thank you for pointing this out. We have corrected them.

Proteomics: Please provide a searchable excel file listing all significantly up or down regulated proteins, columns with respective p-values/fold change, and a column highlighting golgi-resident proteins.

Response: The MS proteomics data have been deposited to the ProteomeXchange consortium via the jPOST partner repository with the dataset identifier PXD049220. We have also provided a excel file of the data.

'We did not observe the fragmentation of Golgi in HeLa cells during starvation; however, with bafilomycin A1 (Baf A1), a vacuolar ATPase inhibitor, we observed the significant appearance of punctate structures of EGFP-YIPF3 and EGFP-YIPF4 in the cytoplasm, in addition to the ribbon-like Golgi structure (Fig 2A and B).'

1. The logic (start of the paragraph) as well as figure legends indicate that experiments were performed in MEFs, not HeLa. Please comment / correct.

Response: We apologize for the mistake. We corrected HeLa to MEFs.

2. While images are convincing regarding the general message (more than in growing conditions), number of puncta in Fig. 2A/C (EBSS+Baf) seem to be higher than in the bar quantification (2B). Was counting done manually or semi-automatically (automated background, automated numbers /cell)? Can this be presented more intuitively (e.g. adding an image with a mask in EV Fig so that reader understands which dots were counted and which not and why)?

Response: We have replaced more representative images. We counted the puncta manually using FIJI as showing below.

The processing of Halo-mGFP-YIPF3 or Halo-mGFP-YIPF4 was suppressed by the deletion of FIP200, showing that YIPF3 and YIPF4 were degraded by autophagy upon starvation (Fig 3 D-G). '□ change to 'Increased processing ... upon starvation...' Clearly, constructs were processed also in FIP200 KO to some extent.

Response: We have changed the phrase as the reviewer suggested.

Fig 4B & text: based on the provided data, endogenous YIPF3 did NOT bind to FLAG-tagged LC3B, but only to FLAG-tagged GABARAP and GABARAPL1. Please correct. Endogenous LC3B did seem to interact with FLAG-tagged YIPF3. Maybe a point for discussion. Interesting also in respect of recent finding that N-term of LC3B may interact with membrane (Tooze lab, PMID: 37288820).

Response: We observed the binding of YIPF3 with LC3B but it was very weak in the previous figure. Therefore, we redid the experiment to obtain more clear results. The interactions of YIPF3 with LC3B, GABARAP, and GABARAPL1 have shown in Fig. 4B. This is consistent with the results in Fig.4C showing that endogenous LC3 binds to FLAG-YIPF3.

Fig EV3: Reminds a bit on ER - please add an ER marker in parallel (for LIR mutant) to see the impact / correlation.

Response: As the reviewer suggested, we have generated MEFs stably expressing mCherry-SEC61 and EGFP-YIPF3 WT (or LIR mutant) and observed whether the expression of YIPF3 LIR affects the ER. YIPF3 LIR mutant did not localized to the ER and

did not affect its morphology. We added the data in Fig. EV3.

Fig 5C & text: since the authors used purple to visualize the mRFP part, there are no red-only puncta. Please change or adapt the text to avoid confusions. Based on the provided images, KO of YIPF3,4 does change the morphology upon starvation (similar to FIP200 KO). Please clarify.

Response: We thank the reviewer for the suggestion. We have changed “red-only puncta” to “mRFP-only puncta (mRFP⁺, GFP⁻) “. We have performed IF using anti-GM130 and anti-p230 antibodies to visualize the Golgi, and observed that the Golgi morphology was not changed in YIPF-KO cells. We added the data in Fig. EV1.

Fig 6B, 4th lane, Halo blot: The quantification shows a drop in YIPF3 KO cells. However, in contrast to the WT in lane 1, there is a clearly visible Halo signal in YIPF3 KO cells. As I understood the assay, this would suggest a higher activity in this specific experiment. Please explain.

Response: For the quantification of Golgiphagy activity, the cleaved Halo-tag band was divided by the total Halo-tag band [Halo/(Halo-mGFP-Golgi+Halo)]. Therefore, if the expression levels of the probes between cells are different, this causes the unfavorable impression that there was no correlation between the appearance and the quantitative results. To remedy this problem, we sorted cells expressing the same level of the probes and redid the experiments using these cells. In the revised manuscript, we were able to show the data that matched the appearance and the quantitative results.

Fig 6B: FIP200 levels strongly dropped in starved YIPF4 KO cells. Was this consistent in all experiments?

Response: We redid the experiment and confirmed that FIP200 levels were not changed by deletion of YIPF4.

Fig.6E: change label siYIPF4 for siYIPF3/4

Response: We thank the reviewer for the correction. We have changed the label.

'... (Fig 6B). This confirmed that the appearance of free Halo tag was dependent on autophagy and lysosomal degradation. These findings showed that the mRFP-EGFP-Golgi reporter can be used to measure Golgiphagy flux quantitatively by SDS-PAGE.' □ I assume the authors meant Halo-mGFP-Golgi reporter.

Response: We thank the reviewer for the correction. We have changed “mRFP-EGFP-Golgi” to “Halo-mGFP-Golgi”.

Text referring to Fig 6B,C: the note (equivalent to YIPF3/4 DKO) is inconsistently used in comparison to previous paragraphs and therefore implies that this specific experiment was also done in a DKO line.

Response: We have removed “(equivalent to YIPF3/4 DKO)” to avoid confusion.

'Indeed, CALCOCO1 was recently reported to be a soluble Golgiphagy adaptor.' Please change to receptor/co-receptor for consistency and add the respective reference.

Response: We have change “adaptor” to “receptor/co-receptor”, and added the reference.

7A, B: The quantification of specifically YIPF3 levels presented, do not seem to correlate with the presented Immuno blot (YIPF3 antibody). I understood that the total of all YIPF3 signals (endogenous + exogenous) were taken into account. However, already the signal of HA-YIPF3 in lane 3,4 (by eye) surely more than compensates the decrease of the endogenous protein. And in lane 7,8 a 3-fold increase seems far too little. Maybe for the stabilization of the endogenous protein. The signal of the exogenous protein seems to be 10x more. Please reconfirm the given quantification and potentially correct fig & text.

Response: We reconfirmed the quantification, and it was correct. In the previous manuscript, the bands of Flag were so dense that it gave the wrong impression. We redid the experiment several times to confirmed the quantification and have showed a representative data in the revised manuscript.

As this is the only overexpression condition where Golgiphagy is increased, please provide sample images of the Golgi in Ext Fig. for YIPF3 / YIPF4 co-expressing cells in extended figures.

Response: We added the images of the Golgi in YIPF3/YIPF4 co-expressing cells. Due to the limitation of the number of Extended figures, we added the data in Appendix Figure S1.

Fig8: By eye, the Halo signal in lane 6 seems to be stronger than in lane 2, which would indicate a small overcompensation. This is not visible in the quantification, which show a non-significant reduction of activity compared to WT cells. While of no relevance for the message of the paper, it is irritating that blot and quantification does not seem to correlate.

Response: For the quantification of Golgiphagy activity, the cleaved Halo-tag band was divided by the total Halo-tag band [Halo/(Halo-mGFP-Golgi+Halo)]. Therefore, if the expression levels of the probes between cells are different, this causes the unfavorable impression that there was no correlation between the appearance and the quantitative results. To remedy this problem, we sorted cells expressing the same level of the probes

and redid the experiments using these cells. In the revised manuscript, we were able to show the data that matched the appearance and the quantitative results.

Non-essential suggestions:

Fig 3B - squares and circles were difficult to identify in the printed pdf. Please use more different forms, e.g. empty triangle and filled circle.

Response: Thank you for the suggestion. We have rewritten the graphs using filled circle and empty square.

Fig 8A: I would suggest to remove the labels on the left side of the YIPF3 blot. Form I and III of HA-YIPF3 runs on too similar height as form II and I of the endogenous protein, respectively. The info is not essential and on the first sight one gets the impression that there is suddenly remaining endogenous protein in the KO cells in lane 5-8.

Response: Following the reviewer's suggestion, we have removed the labels on the left side of the YIPF3 blot.

Given the previous report, the finding of the authors that CALCOCO1 KD did not decrease Golgiphagy in HeLa cells under basal as well as starvation is unexpected. Please comment on this contradictory finding.

Response: We thank the reviewer for raising the important point. Knockdown of CALCOCO1 did not decrease Golgiphagy judging from our Halo-mGFP-Golgi processing assay (Fig 6D and E). Very recently Harper's group published a related paper and reported that proteomic analysis showed that CALCOCO-KO cells exhibited an extent of Golgi membrane protein turnover comparable to that of control cells (Hickey et al. *Nature*, 2023, <https://doi.org/10.1038/s41586-023-06657-6>). These results indicate that the YIPF3-YIPF4 complex may contribute to Golgiphagy more than CALCOCO1. We added this in the discussion section (page 15, line 346).

Referee #3:

Kitta et al report that the two five-transmembrane proteins YIPF3 and -4 located in the cis-Golgi act together as a Golgiphagy receptor. YIPF3 binds to ATG8 family proteins via a LIR motif which is regulated by phosphorylation. YIPF3 and -4 form a complex. YIPF4 does not bind to ATG8s but is required for the stability of YIPF3 so that these two transmembrane proteins together act as a Golgiphagy receptor upon starvation dependent on the LIR motif in YIPF3. They become degraded in a Bafilomycin A1-sensitive manner upon starvation and Halo-tagged YIPF3 or -YIPF4 constructs accumulate in the lysosome upon starvation. The authors also developed a general Golgiphagy reporter as a tandem tag and as a Halo-tag

construct containing a 10-amino acid sequence shown earlier to act as a Golgi targeting motif. Using this reporter and KO cells they show that the YIFP3/-4 complex is required for efficient Golgiphagy upon starvation. Overexpression of the complex also increased Golgiphagy while reconstitution of KO cells with a YIFP3 LIR mutant could not rescue starvation induced Golgiphagy. Expression of the YIFP3 LIR mutant showed an elongated Golgi morphology suggesting that YIFP3/-4-mediated Golgiphagy is involved in maintaining normal Golgi morphology. The authors show that YIFP3 is phosphorylated on serine using a pSer Ab following IP of FLAG-tagged YIFP3 from cells. Immediately N-terminal to the core LIR motif (FEEDM) are two Ser residues and mutation of these to Ala residues inhibited binding of LC3 to YIFP3 and did also inhibit starvation induced Golgiphagy.

This is a very interesting study of high quality reporting the first membrane-spanning Golgiphagy receptor complex and also specific reporters for monitoring Golgiphagy. The study is very well performed with good controls yielding convincing results, and where relevant quantifications of results have been performed.

I have only one major critical point to make that would strengthen the story if addressed.

1. The authors claim that phosphorylation of the LIR motif in YIFP3 is required for YIFP3-ATG8 interaction and for YIFP3-YIFP4-mediated Golgiphagy.

a) The authors show that YIFP3 can be phosphorylated on serine, but it is not shown that the S45 and/or S46 residues are phosphorylated. A control where the S45A/S46A mutant YIFP3 is immunoprecipitated after expression in YIFP3 KO cells and blotted with pSer Ab is lacking from Fig 9B. Did the authors try this experiment? It is highly likely that other sites in YIFP3 is phosphorylated so that the result would still be that a band is seen with the pSer Ab, but if the band disappears or is weakened significantly it would be suggestive of these sites being targets of phosphorylation. Ideally, it would be important to see if phospho-proteomics experiments would show if S45/S46 are phosphorylated and if this is induced upon starvation.

Response: We thank the reviewer for raising the important point. As the reviewer suggested, we have performed pSer-IP and observed clear decrease of phosphorylation in YIFP3 S45A/S46A mutant proteins, suggesting Ser45/Ser46 are targets of phosphorylation. We added the data in Fig.9B. We have also performed phospho-proteomics experiments to show directly if Ser45/Ser46 are phosphorylated, but we could not detect the phosphorylation. The pSer-IP results are very suggestive but still indirect, therefore, we have tone down our claim about the regulation of YIFP3 by phosphorylation in the revised manuscript.

b) The authors could test Asp or Glu mutants of S45 and S46 and test if these induce Golgiphagy when expressed in YIFP3 KO cells and also do binding experiments to show if the S45D/S46D mutant (or E mutants) binds more strongly to the ATG8s in pull down assays or IPs than the WT YIFP3.

Response: As the reviewer suggested, we have tested if the S45D/S46D mutant and S45E/S46E mutant bind more strongly to the ATG8s than the WT YIPF3 by IP. The phosphomimetic YIPF3 SE and SE mutants showed stronger affinity to LC3 compared with YIPF3 SA mutants, but less affinity than YIPF3 WT. The result is consistent with ER-phagy receptor TEX264, which cannot be complemented by Asp or Glu mutations (Chino et al., EMBO rep 2022, <https://doi.org/10.15252/embr.202254801>). We added the data in Fig EV5.

c) Do the authors have any clue which protein kinases(s) are involved in phosphorylating S45 and/or S46. This could be tested using chemical inhibitors of some relevant kinases. Exploring the phosphorylation part more will strengthen this paper. If this is not done, the authors need to moderate their conclusions regarding the role of phosphorylation here and discuss this in the Discussion section.

Response:

We agree that identification of kinases involved is important, however, it will be a lot of work and we would like to address this issue in the future. Therefore, we have tone down our conclusions regarding the role of phosphorylation of YIPF3 in the revised manuscript (describe as putative phosphorylation sites). We also discuss this in the Discussion section (page 16, line 351).

Minor point:

No binding to LC3s is seen in the IP in Figure 4B while a strong binding is seen to GABARAPL1 and a weaker binding to GABARAP. However, in Figure 4C LC3 is IPed very efficiently it seems. How do the authors explain this?

Response: We observed the binding of YIPF3 with LC3B but it was very weak in the previous figure. Therefore, we redid the experiment to obtain more clear results. The interactions of YIPF3 with LC3B, GABARAP, and GABARAPL1 have shown in Fig. 4B.

Also, the input of the different FLAG-tagged ATG8as vary a lot in Figure 4B.

It is also unexpected that GABARAP binds seemingly much more poorly than GABARAPL1 as these are very similar and usually show similar binding to a multitude of LIR-containing proteins. Why this difference here. It would be nice to see this experiment repeated with a more equal input and some quantification of the binding.

Response: We cannot think of a good explanation for why GABARAPL1 binds to YIPF3 more than GABARAP, but this result was reproducible. Therefore, we would like to keep this result qualitative rather than quantitative statements.

Dear Akiko,

Thank you submitting a revised version of your manuscript. It was sent to the same three reviewers that originally appraised your work; their comments are attached to the bottom of this email. As you will see, all three referees are satisfied with the changes you made. Please address the remaining few minor recommendations that are listed in these re-review reports. Before we can move forwards towards publication of your manuscript, there are some remaining editorial points which need to be addressed. In this regard, would you please:

- include five keywords,
- remove the AC/CrediT section from the text,
- include callouts in the manuscript text for Figure EV5 and Table EV1,
- upload high-resolution figures individually as separate files,
- upload the Appendix in PDF format; Appendix Figure S1 should be included in this Appendix PDF and listed in a preceding table of contents with page numbers,
- ensure that dataset PXD049220 is made publically available after final acceptance of you manuscript,
- include a legend for figure EV 2d,
- define the annotated p values ****/**/*/* in the legends of figures 2b; 3e, g; 5c; 6c, e; 7b-c; 8b; 9f; EV 4b, d,
- indicate the statistical test used for data analysis in the legends of figures 1a; 2b; 3e, g; 5c; 6c, e; 7b-c; 8b; 9f; EV 4b, d, and
- although 'n' is provided, please describe the nature of entity for 'n' in the legends of figures 2b; 3b, e, g; 5c; 6c, e; 7b-c; 8b; 9f; EV 4b, d.

My colleague Hannah Sonntag will contact you about providing the source data which will be necessary for publication of this work.

Best wishes,

William

William Teale, PhD
Editor
The EMBO Journal
w.teale@embojournal.org

- a point-by-point response to the referees' comments, with a detailed description of the changes made (as a word file).
- a word file of the manuscript text.

- individual production quality figure files (one file per figure)

- a complete author checklist, which you can download from our author guidelines

(<https://www.embopress.org/page/journal/14602075/authorguide>).

- Expanded View files (replacing Supplementary Information)

We realize that it is difficult to revise to a specific deadline. In the interest of protecting the conceptual advance provided by the work, we recommend a revision within 3 months (12th Jun 2024). Please discuss the revision progress ahead of this time with the editor if you require more time to complete the revisions. Use the link below to submit your revision:

Referee #1:

The revised manuscript by Kitta et al has improved a lot since the previous version, with better explanations (although some of these are only seen in the response) and toning down previous overinterpretations. While the work still leaves important questions open (e.g., regarding the suggested phosphorylation of YIPF3 and the identity of kinases responsible), this version may be suitable for publication.

Comments:

1. Affiliation: Faculty of Life Sciences, Kyoto Sangyo University is mistakenly marked as 4 instead of 5
2. Colocalization of YIPF3 with cis and medial Golgi markers is a welcome addition, but text only highlights this in starved Baf treated cells. Please add the extent of colocalization in untreated cells (also shown in Fig EV2 panel D).
3. Please double-check for mistakes. For example, Fig EV2 contains panels A-D while its legend only describes panels A-C.

Referee #2:

The authors have addressed all my concerns sufficiently and I congratulate them on their nice piece of work.

As a last minor point, I would suggest to add another column to their excel file after p-value and log-change indicating statistically significant changes by a '+'. It makes it much easier for non-experienced readers/students to grasp the data.

Referee #3:

The authors have done a very nice job revising their manuscript.
The new data have very good quality.

I have only two minor points:

Quantification of colocalization is lacking for the confocal microscopy images in Figures 2C, 2E, and EV2B.
In line 313 "EV3" needs to be corrected to "EV5"

Response to the reviewer comments:

First of all, we would like to thank the reviewers for the positive decision on our revised manuscript. We also appreciated the helpful and constructive comments on it. We have revised our manuscript according to all the requests from the reviewers. The revisions are noted in colored letters in the manuscript. We hope that our responses are reasonable and acceptable for the reviewers.

Referee #1:

The revised manuscript by Kitta et al has improved a lot since the previous version, with better explanations (although some of these are only seen in the response) and toning down previous overinterpretations. While the work still leaves important questions open (e.g., regarding the suggested phosphorylation of YIPF3 and the identity of kinases responsible), this version may be suitable for publication.

Comments:

1. Affiliation: Faculty of Life Sciences, Kyoto Sangyo University is mistakenly marked as 4 instead of 5

Response: We are sorry for our careless mistakes in the manuscript. We have corrected it.

2. Colocalization of YIPF3 with cis and medial Golgi markers is a welcome addition, but text only highlights this in starved Baf treated cells. Please add the extent of colocalization in untreated cells (also shown in Fig EV2 panel D).

Response: We would like to thank the reviewer for pointing this out. We have described the colocalizations in both untreated and starved cells (in page 6, line 128).

3. Please double-check for mistakes. For example, Fig EV2 contains panels A-D while its legend only describes panels A-C.

Response: Thank you for pointing out the mistake. We have corrected the legend of FigEV2D. The manuscript has been rechecked and errors corrected.

Referee #2:

The authors have addressed all my concerns sufficiently and I congratulate them on their nice piece of work.

As a last minor point, I would suggest to add another column to their excel file after p-value

and log-change indicating statistically significant changes by a '+'. It makes it much easier for non-experienced readers/students to grasp the data.

Response: We would like to thank the reviewer for this suggestion. We have revised Table EV1 as suggested.

Referee #3:

The authors have done a very nice job revising their manuscript.
The new data have very good quality.

I have only two minor points:

Quantification of colocalization is lacking for the confocal microscopy images in Figures 2C, 2E, and EV2B.

Response: According to the reviewer's suggestion, we have quantified the colocalization in Fig2C, E and EV2D and added the quantitative values in the text of the corresponding parts (page 6, line 115, 124-125, 128-129).

Only for Fig2EVB, it was difficult to quantify. BafA1 caused morphological changes of lysosomes, this made it difficult to quantify the colocalization between YIPFs and LAMP1. Therefore, we would like to remain the description qualitative rather than quantitative in the manuscript (page 6, line 115-116).

In line 313 "EV3" needs to be corrected to "EV5"

Response: We are sorry for our careless mistakes in the manuscript. We have corrected EV3 to EV5.

Dear Akiko,

I am pleased to inform you that your manuscript has been accepted for publication in the EMBO Journal.

Congratulations! This is a really insightful series of experiments.

Best wishes,

William

William Teale, PhD
Editor
The EMBO Journal
w.teale@embojournal.org
